# Immunomodulatory, Antioxidant, and Potential Anticancer Activity of the Polysaccharides of the Fungus *Fomitiporia chilensis*

**DOI:** 10.3390/molecules29153628

**Published:** 2024-07-31

**Authors:** Roberto T. Abdala-Díaz, Virginia Casas-Arrojo, Pablo Castro-Varela, Cristian Riquelme, Paloma Carrillo, Miguel Ángel Medina, Casimiro Cárdenas, José Becerra, Claudia Pérez Manríquez

**Affiliations:** 1Department of Ecology and Geology, Institute of Blue Biotechnology and Development (IBYDA), Malaga University, E-29071 Malaga, Spain; abdala@uma.es (R.T.A.-D.); virginiac@uma.es (V.C.-A.); 2FICOLAB Microalgal Research Group, Department of Botany, Faculty of Natural and Oceanographic Sciences, University of Concepción, Concepción PC 304000, Chile; pabcastro@udec.cl; 3Mycology Laboratory, Institute of Biochemistry and Microbiology, Universidad Austral de Chile, Isla Teja, PO 567, Valdivia PC 5049000, Chile; cristian.riquelme02@alumnos.uach.cl; 4Department of Molecular Biology and Biochemistry, Faculty of Sciences, University of Málaga, Andalucía Tech, E-29071 Málaga, Spain; pcarrillo@uma.es (P.C.); medina@uma.es (M.Á.M.); ccg@uma.es (C.C.); 5Malaga Biomedical Research Institute and Nanomedicine Platform (IBIMA PlataformaBIONAND), C/Severo Ochoa, 35, E-29590 Málaga, Spain; 6Network Biomedical Research Center for Rare Diseases (CIBERER), U741, E-28029 Málaga, Spain; 7Research Support Central Services (SCAI) of the University of Málaga, E-29071 Málaga, Spain; 8Laboratory of Chemistry of Natural Products, Department of Botany, Faculty of Natural and Oceanographic Sciences, University of Concepción, Concepción PC 304000, Chile; jbecerra@udec.cl; 9Technological Development Unit, University of Concepción, Concepción PC 304000, Chile

**Keywords:** *Fomitiporia chilensis*, diabetic cardiomyopathy, NF-κB inhibitors, proinflammatory cytokines, proteomeXchange

## Abstract

*Fomitiporia* species have aroused the interest of numerous investigations that reveal their biological activity and medicinal potential. The present investigation shows the antioxidant, anticancer, and immunomodulatory activity of acidic polysaccharides obtained from the fungus *Fomitiporia chilensis*. The acidic polysaccharides were obtained for acidic precipitation with 2% O-N-cetylpyridinium bromide. Chemical analysis was performed using FT-IR and GC-MS methods. The antioxidant capacity of acidic polysaccharides from *F. chilensis* was evaluated by scavenging free radicals with an ABTS assay. Macrophage proliferation and cytokine production assays were used to determine the immunomodulatory capacity of the polysaccharides. Anti-tumor and cytotoxicity activity was evaluated with an MTT assay in the U-937, HTC-116, and HGF-1 cell lines. The effect of polysaccharides on the cell cycle of the HCT-116 cell line was determined for flow cytometry. Fourier Transform-infrared characterization revealed characteristic absorption peaks for polysaccharides, whereas the GC-MS analysis detected three peaks corresponding to D-galactose, galacturonic acid, and D-glucose. The secreted TNF-α concentration was increased when the cell was treated with 2 mg mL^−1^ polysaccharides, whereas the IL-6 concentration was increased with all of the evaluated polysaccharide concentrations. A cell cycle analysis of HTC-116 treated with polysaccharides evidenced that the acidic polysaccharides from *F. chilensis* induce an increase in the G0/G1 cell cycle phase, increasing the apoptotic cell percentage. Results from a proteomic analysis suggest that some of the molecular mechanisms involved in their antioxidant and cellular detoxifying effects and justify their traditional use in heart diseases. Proteomic data are available through ProteomeXchange under identifier PXD048361. The study on acidic polysaccharides from *F. chilensis* has unveiled their diverse biological activities, including antioxidant, anticancer, and immunomodulatory effects. These findings underscore the promising therapeutic applications of acidic polysaccharides from *F. chilensis*, warranting further pharmaceutical and medicinal research exploration.

## 1. Introduction

The genus *Fomitiporia*, initially described by Murrill, belongs to the Hymenochaetaceae Donk family (Hymenochaetales, Agaricomycetes Basidiomycota) and comprises a varied and complex group of polypore fungi with worldwide distribution [1,2]. The species in this genus inhabit the wood of numerous forest species and act primarily as pathogens, inducing white rot and canker [2,3]. Nevertheless, certain mushrooms within this genus have potential medicinal properties and are used in traditional medicine to treat heart diseases and gastrointestinal cancer [4].

*Fomitiporia chilensis* Rajchenb. & Pildain is a species with a perennial basidioma of irregular shape. It typically grows on stumps, fallen branches, and dead tissue of *Peumus boldus* (boldo) and *Cryptocarya alba* (peumo), causing white and fibrous wood rot [1]. The species’ distribution is constrained by its host plants, resulting in endemism limited to the sclerophyllous forests of central Chile. This area is characterized by its resistant vegetation adapted to the drought conditions of the region’s Mediterranean climate, and was recognized as one of the 25 biodiversity hotspots [5]. In recent years, *Fomitiporia* species have aroused the interest of numerous investigations focused on their taxonomic and phylogenetic position [1,3]; however, studies that reveal its biological activity and medicinal potential are scarce. Triterpenes extracted from a specimen of the species *Fomitiporia aethiopica* collected in Kenya exhibited notable cytotoxicity against several mammalian cell lines [6]. Specifically, the triterpenes exhibited an effect against MCF-7, A431, and PC-3 with IC_50_ values of 16, 14, and 8 µg mL^−1^, respectively. Previous studies have reported that the total extract of the species *Fomitiporia ellipsoidea*, a polypore from southern China, presents antioxidant components with great activity. The phelligridin K exhibited the highest Trolox equivalent antioxidant capacity (TEAC) against the ABTS radical at a concentration of 3.56 mM of Trolox. In comparison, inoscavin C, and inonoblin B, when tested, exhibited results equivalent to 2.82 and 2.04 mM Trolox, respectively [4]. Additionally, significant antioxidant activity was registered in the G1 fraction of polysaccharides obtained from *Fomitiporia punctata*, evaluated by the increase in inhibition of erythrocyte hemolysis dependent on the concentration of polysaccharide used, reaching up to 73.58% inhibition [7]. Furthermore, this study revealed that concentrations of 250 and 200 µg mL^−1^ of polysaccharide led to 51% and 68.57% absorption of the DPPH and superoxide radicals, respectively. The activities of fungi that are immunomodulatory, antioxidant, antitumor, anti-inflammatory, and antiviral are primarily linked to their polysaccharide content and structural characteristics [7,8]. Accordingly, high molecular weight polysaccharides can directly activate leukocytes and stimulate their phagocytic and cytotoxic capacity [8]. 

*Fomitiporia* species exhibit a wide range of biological activities, from cytotoxicity to antioxidant and immunomodulatory effects, highlighting their potential for medicinal uses. Understanding the mechanisms behind these activities could lead to developing new therapeutic interventions using the bioactive compounds found in these fungi. This research aimed to determine the antioxidant, anticancer, and immunomodulatory activity of acidic polysaccharides obtained from the fungus *F. chilensis.* Through a comprehensive analysis involving various assays and studies, the objective is to evaluate the therapeutic potential of these acidic polysaccharides and contribute to understanding their biological effects, particularly in the context of potential medicinal applications.

## 2. Results

### 2.1. Authentication of Basidiocarps

The identity of the basidiome used in this research was confirmed based on macro and micromorphological features, compared with the holotype and other numerous specimens (paratypes) of the species originally described from the same locality and through its host relationship [1], and because *F. chilensis* is the single species of *Fomitiporia* recorded in Chile [9].

### 2.2. Polysaccharides Characterization/Chemical Assessment

#### 2.2.1. Total Carbon (C), Total Hydrogen (H), Total Nitrogen (N), Total Sulfur (S), and Total Protein

In the present study of *F. chilensis*, an elemental analysis was performed on the extracted fungal polysaccharides. *Fomitiporia chilensis* polysaccharides (FcPs) displayed a high percentage of carbon (40.03%), followed by nitrogen (2.14%), hydrogen (6.08%), and sulfur (0.97%). Table 1 shows the elemental analysis of FcPs and protein content (Table 1).

Accordingly, the obtained molar (C/N ratio) value was 21.8 in FcPs. 

#### 2.2.2. Fourier Transform Infrared Spectroscopy (FT-IR) 

The FT-IR spectra in the wavenumber range of 400 cm^−1^ to 4000 cm^−1^ were utilized for data analysis. The FT-IR data indicated that the polysaccharides exhibit typical carbohydrate patterns, allowing for the identification of specific functional groups and molecular arrangements (Figure 1). 

#### 2.2.3. Gas Chromatography-Mass Spectrometry (GC-MS)

Based on the GC-MS sugar analysis method, the monosaccharides identified in FcPs are presented in Figure 2. Four main peaks were detected at 27.77, 28.62, 28.87, and 29.28 min, integrated to 53.02%, 16.02%, 6.83%, and 6.27%, respectively. Analysis of the mass fragmentations confirmed the assignation of the peaks to galactose (Gal), galacturonic acid (GalA), and glucose (Glc) isomer α and β, respectively, indicating the presence of a galactan containing lateral chains formed by galacturonic acid and glucose (Table 2).

### 2.3. Biological Assessment 

#### 2.3.1. Antioxidant Activity (ABTS Method)

Antioxidant activity was evaluated in different concentrations of polysaccharide from *F. chilensis*. In the total antioxidant capacity (expressed as a percentage of the radical scavenging capacity), all samples showed activity (Figure 3). The concentration at 25 µg mL^−1^ and 50 µg mL^−1^ polysaccharide showed the lowest activity with 6.5 ± 0.3% and 6.99 ± 0.5% of scavenging effects, respectively. Moreover, the high concentration at 500 µg mL^−1^ of the polysaccharide had the greatest activities, with approximately 23.55 ± 0.9% scavenging effects. Significant differences (*p* ≤ 0.05) were observed between the antioxidant activity of concentrations of polysaccharides (Figure 3). 

#### 2.3.2. Cell Viability of Lines RAW 264.7, U-937, HTC-116, and HGF-1 

The cytotoxic effects of the FcPs on RAW 264.7, U-937, HTC-116, and HGF-1 cells were determined using the MTT assay. After 72 h of incubation, the IC_50_ values were calculated at a maximum concentration of 100 µg mL^−1^. The IC_50_ values for each cell line were as follows: RAW 264.7 (598.83 µg mL^−1^) (Figure 4a), U-937 (3177.09 µg mL^−1^) (Figure 4b), and HTC-116 (1154.63 µg mL^−1^) (Figure 4c). However, the IC_50_ for HGF-1 could not be determined, since this value was not reached at the polysaccharide concentrations studied (shown in Figure 4d). The MTT assay results indicated that the tested polysaccharides inhibited cell proliferation in a dose-dependent manner for RAW 264.7, U-937, and HTC-116 cell lines. This study demonstrated that the FcPs induced potent cytotoxicity in RAW 264.7, U-937, and HCT-116 cells. Among the cancer cell lines tested, RAW 264.7 and HCT-116 showed the highest susceptibility to treatment with the FcPs, resulting in increased growth inhibition. 

#### 2.3.3. Determination of Cytokines (IL-6 and TNF-α) in RAW 264.7 and THP-1 Cell Line

This trial evaluated the production of proinflammatory cytokines IL-6 and TNF-α. For this purpose, the RAW 264.7 mouse macrophage and THP-1 human cell lines were used with the following polysaccharide concentrations: 1 μg mL^−1^, 5 μg mL^−1^, 25 μg mL^−1^, 50 μg mL^−1^, 75 μg mL^−1^, and 100 μg mL^−1^ for IL-6 and 0.1–2.5 mg mL^−1^ for TNF-α, respectively. For the positive control, *E. coli* lipopolysaccharide (LPS) was used at a concentration of 0.5 μg mL^−1^. The maximum concentration of IL-6 obtained within the concentrations studied in RAW 264.7 cells was 7132 ± 552 pg mL^−1^ when a concentration of polysaccharides of 25–100 μg mL^−1^ (Figure 5a) was applied. In the case of the production of TNF-α by THP-1 cells, the maximum was 9028 pg mL^−1^ at the concentration of 2.0 mg mL^−1^ (Figure 5b). Both Figures show that for IL-6 and TNF-α, the synthesis and accumulation by both cell lines increased as a higher concentration of FcPs was applied. Specifically, IL-6 synthesis increased from 4244 ± 869 to 7209 ± 484 pg mL^−1^ as it passed a polysaccharide concentration of 5 to 25 μg mL^−1^. Similarly, TNF-α production increased from 2124 ± 1250 to 9028 ± 3590 pg mL^−1^ when the polysaccharide concentration increased from 1.0 to 2.0 mg mL^−1^. However, in none of the cases did the cytokine levels become saturated.

#### 2.3.4. HCT116 Cell Cycle Analysis Using Flow Cytometry

Cell cycle analysis was conducted on HCT116 cells treated with 0.5, 1.0, or 2.0 mg mL^−1^ of FcPs. The analysis revealed that these polysaccharides induced a significant increase in Sub G0/G1 phase with a 10.4 ± 1.4%, 14.5 ± 3.2%, and 16.6 ± 4.2% apoptotic population (at 0.5, 1.0, and 2.0 mg mL^−1^, respectively) induced by these polysaccharides, indicating that it is driven HCT116 cells towards apoptosis. The increase in apoptotic population was accompanied by a concomitant decrease in the G2/M phase, as depicted in Figure 6.

#### 2.3.5. Proteomic Analysis in HGF-1 Cells

The proteomic analysis of HGF-1 cells allowed for the high-confidence identification of 1877 proteins (<1% false discovery rate). Table 3 and Figure 7 present the proteins significantly deregulated post-treatment with FcPs (Log2 fold-change > 0.5, *p*-value < 0.05).

### 2.4. STRING Analysis of Protein Networks

A protein-protein interaction networks functional enrichment analysis of the significantly deregulated proteins was carried out using STRING (Search Tool for the Retrieval of Interacting Genes/Proteins). The STRING database assembles information about both the known and predicted protein-protein interactions based on numerous sources, including experimental repositories, computational prediction methods, and public text collections. This analysis revealed that post-treatment differentially expressed proteins have functional associations and are involved in different biological processes, as shown in Figure 8, mainly related to viral life cycle, viral processes, host-virus interaction, response to cytokine, and TGF-beta signaling pathways. In addition, we used the STRING database to look for known functional interactions among the significantly deregulated proteins and the key immunological markers IL-6 and TNF-α (Figure 9), whose expressions were induced when TPH1 cells were treated with polysaccharides from *F. chilensis*, as shown by the results obtained using ELISA. Several deregulated proteins functionally interact with IL-6 and TNF-α (see Table Inside Figure 9), more of them were over-expressed (eight), and only one was under-expressed.

## 3. Discussion

### 3.1. Polysaccharides Characterization

Polysaccharide extraction from *F. chilensis* was carried out using an ethanol-based method to eliminate the fungi’s fat-soluble components, followed by hot-water extraction and cetavlon to precipitate the acidic polysaccharides selectively. The polysaccharide yield from *F. chilensis* was 0.26% using this method. A study conducted on polysaccharide extraction from *Ganoderma lucidum* found that hot-water extraction resulted in a yield of 8.32%, which was higher than yields obtained from other methods such as ultrasonic-assisted extraction (6.87%) and microwave-assisted extraction (7.21%) [10]. Another study reported that, through hot-water extraction, the yield of polysaccharides was 11.92% for *Ganoderma*, and the yield of polysaccharides through cetavlon precipitation was 3.34% [11]. A separate investigation also found that the yield of polysaccharides by hot-water extraction was 8.32%, and the yield of polysaccharides by cetavlon precipitation was 2.78% [10]. Regarding *Pleurotus*, a study reported that the yield of polysaccharides by hot-water extraction was 18.11%, and that of polysaccharides by cetavlon precipitation was 6.04% [12]. It can be observed that the quantity of polysaccharide extracted from *Ganoderma* and *Pleurotus* through the methods of hot-water extraction and cetavlon is dependent on the extraction conditions, such as temperature, time, pH, and solvent/solid ratio. Despite this, the polysaccharide yield from hot-water extraction generally surpasses that of cetavlon precipitation, and the yield of polysaccharides obtained from *Pleurotus* is typically more substantial than that from *Ganoderma*. Various fungi adjust the quantity of the polysaccharides they produce based on the quality of their interplay with their hosts. This interaction could manifest in different ways with their hosts, namely, as parasitic, symbiotic, or saprophytic. As such, one could speculate that the observed yield of polysaccharides in *F. chilensis* is an adaptive response to the specific environmental factors in which they exist.

#### 3.1.1. Total Carbon (C), Total Hydrogen (H), Total Nitrogen (N), Total Sulfur (S), and Total Protein

The elemental composition of polysaccharides, encompassing carbon, hydrogen, nitrogen, and sulfur, is crucial for understanding their chemical structure and potential biological activities. Analyzing these elements provides valuable insights into the properties and functions of polysaccharides, aiding in establishing structure-function relationships and identifying possible applications in various fields, including medicine, nutrition, and biotechnology [13,14]. The present study of the extracted polysaccharides found that the carbon, hydrogen, nitrogen, and sulfur content contributed 40.03%, 6.08%, 2.14%, and 0.97%, respectively (Table 1). The percentages of C, H, N, and S were determined in the purification fraction of the crude polysaccharides of *Pleurotus eus* (Berk.) Saccs on a DEAE-52 cellulose ion exchange column were 41.08%, 6.25%, 4.99%, and 2.79%, respectively [15]. The elemental composition of the polysaccharide fraction isolated from the basidiocarps of the cultivated medicinal mushroom *Pleurotus ostreatus* was analyzed [16]. The carbon content ranges from 39.7 to 41.7%, while the hydrogen range is between 6.7% and 6.8%. Then, nitrogen is between 1.6 and 4.4%. Finally, sulfur was present in all fractions at a percentage lower than 1%, ranging from 0.1 to 0.2%. In a more detailed study, *Agaricus brasiliensis* cell wall polysaccharides isolated from the fruiting body (F.R.), and mycelium (MI) and their respective sulfated derivatives (FR-S and MI-S) were chemically characterized using elemental analysis. The carbon, nitrogen, hydrogen, and sulfur content of *A. brasiliensis* fruiting body polysaccharides and *A. brasiliensis* mycelial polysaccharides were as follows (CHN, % *w*/*w*): 34.42, 5.56, 2.50; 32.07, 4.36, and 1.63, respectively [17]. In both samples, the sulfur content was zero. Regarding the results of the analysis of elements, the amount of carbon, hydrogen, nitrogen, and sulfur was comparable to the other polysaccharides (Table 4).

The protein content of polysaccharides is crucial for assessing their nutritional value and health benefits. Analyzing the protein composition of polysaccharides from mushrooms provides insights into their biological activities and uses. The protein content of polysaccharides from FcPs was 18.24%. This correlates with the protein contents of 17.06% reported to *P. ostreatus* [18]. However, the protein content of the fruiting body of *A. brasiliensis*, commonly known as sun mushroom or Brazil mushroom, is higher and varies considerably. Reported values for this mushroom range from 28.9% to 39.2% for dry mushrooms [19]. *F. chilensis*, compared to *P. ostreatus* and *A. brasiliensis*, with its protein content, can be considered a valuable nutritional source with potential health benefits (Table 4).

#### 3.1.2. Fourier Transform Infrared Spectroscopy (FT-IR)

The FT-IR spectra in the wavenumber range of 400 cm^−1^ to 4000 cm^−1^ were utilized for data analysis. The analysis of FT-IR spectra of polysaccharides from mushrooms, such as FcPs, provides valuable insights into their structural characteristics. The FT-IR data indicated that the polysaccharides exhibit typical carbohydrate patterns (Figure 1). Three characteristic strong bands from 3600 to 3000 cm^−1^ point to the presence of the symmetrical(s) and asymmetric(as) axial stretching modes of carbohydrates, and proteins (*ν* (O-H) or *ν* (N-H)), indicating that there are strong intermolecular and intramolecular interactions between polysaccharide chains [20]. The weak peak around 2920 cm^−1^ is due to *ν* (C-H) stretching and bending vibrations [21]. The intense signal near 1630 cm^−1^ may be associated with the symmetrical stretching vibration of the *ν*_as_ (COO-) and amide I *ν* (C=O) groups [22,23]. In addition, the absorption peaks at 1000–1200 cm^−1^ (1157/1072 cm^−1^) corresponded to *ν* (C-O-C) glycosidic ether linkage, *ν* (C-H) anomeric carbon, *ν* (C-O) primary alcohol stretching vibrations and proved that FcPs had pyranose rings [24]. Furthermore, absorption peaks between 1000 and 600 cm^−1^ indicated the presence of β- and α-glycosidic linkages, respectively [25]. After that, it was deduced that the FcPs contained both α- and β- configurations. The presence of β- and α-glycosidic linkages in polysaccharides plays a crucial role in determining their bioactivity. Several studies have emphasized the significance of these glycosidic linkages in influencing the biological properties of polysaccharides. For example, a study conducted by Jen et al. (2021) [26] concluded that polysaccharides with β- (1→3, 1→6) glycosidic linkages exhibited stronger anti-inflammatory activity compared to β- (1→3, 1→4) -linked polysaccharides. This indicates that the specific glycosidic linkage type can impact the bioactivity of polysaccharides, with specific configurations showing enhanced anti-inflammatory effects. Furthermore, the presence of α- and β-glycosidic linkages in mushroom polysaccharides has been associated with discernible patterns in glycosidic bonds and branching sites [27]. This structural feature influences the biological activities of polysaccharides, indicating that the types and combinations of glycosidic linkages are important determinants of bioactivity [28,29]. In conclusion, the presence of β- and α-glycosidic linkages in polysaccharides from mushrooms and other sources is crucial for their bioactivity. The glycosidic linkages influence the anti-inflammatory, antioxidant, immunomodulatory, and other beneficial properties of FcPs, thereby emphasizing the importance of structural features in determining the biological activities of these polysaccharides.

#### 3.1.3. Gas Chromatography Mass Spectrometry (GC-MS)

Analyzing the monosaccharide compositions of polysaccharides from mushrooms is essential for understanding their bioactivity and potential health benefits. The presence of specific monosaccharides, such as glucose (Glc), mannose (Man), galactose (Gal), xylose (Xyl), arabinose (Ara), rhamnose (Rha), and fucose (Fuc), along with galacturonic (GalA) and glucuronic (GlcA) acids, influences the biological properties of these polysaccharides; see Zhao et al. (2020) [30]. Less frequent components of fungal polysaccharides include galacturonic (GalA) and glucuronic (GlcA) acids, as well as monosaccharides such as fructose (Fru) and ribose (Rib). Additionally, amide derivatives of the monosaccharides glucose and galactose, namely N-acetyl-glucosamine (GlcNac) and N-acetyl-galactosamine (GalNac), are present [31]. According to the GC-MS analysis method, the most identified monosaccharides by comparison of the retention times of standard monosaccharides found in the sample were Gal (52.62%), GalA (28.62%), and Glc (28.87%), and other minority monosaccharides. The derivative glucose, galactose, and galacturonic acid have a molecular weight of 482, 482, and 496 gmol^−1^, respectively. These monosaccharides present the same fragments, 73, 133, 147, 204, 217 (*m*/*z*), when ionized by electronic impact at 70 eV, so they can only be identified by comparing their retention times with those of their respective standards. Minor derived monosaccharides would have molecular masses of 380 gmol^−1^ for ribose and xylose and 394 for rhamnose and fucose, respectively. The polysaccharide fraction obtained from the fungus *Fomitiporia punctata* (P. Karst.) Murrill was analyzed using gas chromatography-mass spectrometry (GC-MS) and high-performance liquid chromatography (HPLC). The results showed that it was mainly composed of arabinose, fructose, galactose, and glucose in a molar ratio of 1.6:3.8:19.7:19.7, respectively, the specific molar ratios indicating the relative abundance of each monosaccharide [7]. The monosaccharide compositions of the cold-water-soluble fractions obtained from the basidiomata of *Gymnopilus imperialis* were studied. The two fractions enriched in polysaccharides had high glucose contents (64–65%), suggesting that they were mainly composed of glucans. The main monosaccharide in the third fraction analyzed was galactose (58.8%), indicating the presence of hetero-galactan containing lateral chains formed by mannose (30.7%) and fucose (7.1%) [32]. Furthermore, the extraction method can influence the monosaccharide composition of mushroom polysaccharides. Polysaccharides from two basidiomycete species were extracted using microwave-assisted extraction (MAE) and pressurized liquid extraction (PLE). The results indicate that *Pleurotus ostreatus* and *Ganoderma lucidum* contained 91.0% and 91.3% glucose, and 3.5% and 2.4% galactose, respectively, when extracted using MAE. When extracted using PLE, *Pleurotus ostreatus* and *Ganoderma lucidum* contained 89.6% and 86.5% glucose, and 7.1% and 9.4% galactose, respectively [32]. The presence of glucuronic acid (GlAc) in bioactive mushrooms is significant due to its potential health-promoting properties. Glucuronic acid, a uronic acid derived from glucose, is a key component known for various biological activities and therapeutic benefits [33]. The presence of GlAc was reported by Su et al. (2016) in the edible mushroom *Grifola frondose* in a molar ratio of 5.8:10.5:72.2:7.8:2.2:1.2 (Fuc:Gal:Glc:Man:Rib:GlcAc) [34]. For *Auricularia auricularia*, a medicinal basidiomycete, the presence of GlAc in two polysaccharide fractions was also minor, with a molar ratio of 98.90:0.11:0.38:0.61 (Glc:Man:Gal:GlcAc) and 97.56:0.14:1.04:0.11:0.45:0.50:0.20 (Glc:Man:Gal:Ara:Fuc:GluAc: GalAc) for CEPSN-1 and CEPSN-2, respectively [35]. The molar ratios of GlAc in these mushrooms suggest its presence alongside other monosaccharides. This contributes to the overall bioactivity of the polysaccharides extracted from these species. In conclusion, the monosaccharide compositions of mushroom polysaccharides, including the presence of specific monosaccharides and their ratios, are crucial factors that influence the bioactivity and potential health-promoting properties of these compounds. Understanding the monosaccharide profiles of FcPs is essential for elucidating their biological activities and exploring their therapeutic potential.

### 3.2. Biological Assessment 

#### 3.2.1. Antioxidant Activity (ABTS Method)

The antioxidant potential of polysaccharides extracted from the *F. chilensis* was evaluated using ABTS radicals’ scavenging activity. The antioxidant components of mushrooms include polysaccharides, tocopherols, phenols, carotenoids, ergosterol, and ascorbic acid, among others [36]. Mushroom antioxidants exhibit their protective activity at various stages of the oxidation process, as well as using various mechanisms. The antioxidant activity of mushrooms can be observed at different stages of the oxidation process, and the mechanisms by which they perform said activity are also often varied [37]. The ABTS radical cation formation-based assay is a rapid and efficient method for measuring fungal extracts’ free radical scavenging activities [38].

#### 3.2.2. Cell Viability of Lines RAW 264.7, U-937, HTC-116, and HGF-1

The MTT assay showed toxicity in RAW-264, U-937, and HCT-116 cell lines. The less the maximal effective concentration (IC_50_), the higher the toxicity. The FcPs showed a significantly higher IC_50_ in RAW-264 cells (0.598 mg mL^−1^); meanwhile, the dynamic was the opposite for the HGF-1 (IC_50_ not detected). Carcinogenic cells were also tested to observe possible therapeutic applications. The FcPs showed a high IC_50_ in U-937 (3.17 mg mL^−1^) and a selective index (SI) of 0.18. Similar results were for colon cancer, where the HCT-116 showed IC_50_ of 1.15 mg mL^−1^ and 0.51 of the selective index (SI), suggesting a low antitumor activity against leukemia and colon cancer. The greater SI value, the more selectivity presents the tested compound. According to Indrayanto et al. (2021) [39], compounds with SI values higher than 3 present potential for further investigation due to their toxicity against a cell line. Fungal polysaccharides have shown moderate toxicity in some studies, even though their cellular proliferation and wound-healing properties have been demonstrated [40,41]. The polysaccharide from *F. chilensis* showed similar cellular viability to other compounds extracted from Chilean fungi, such as *Nothophellinus andinopatagonicus* (e.g., acid polysaccharide present concentration of IC_50_ in HCT-116 cells of 1.26 mg mL^−1^ [41]). A previous study also showed that sugars of the polysaccharides could affect the mechanisms directly on tumor cells (Del Corno et al., 2020) [42]. Similar patterns have been demonstrated in algae polysaccharides where galactose monosaccharides are the principal influent [43]. The glucose or mannose in these types of extracts could affect their antitumor and immunomodulatory activities [44]. In this sense, our study is the first approach to this Chilean fungal, which suggests that the extraction procedure could significantly enhance the anticancer activity of polysaccharides, the concentration of the metabolite, and the proportion of the sugars in the chemical structure, as well as the location-related influences of the fungal batch used. Additionally, the medicinal properties can vary enormously under similar conditions depending on the strain, the geographical area, the growing conditions and the substrate used, the part of the mushroom used, and the growing stage at the moment of processing. All these parameters changed the chemical composition of the mushroom and, consequently, its bioactive capacity. Thus, since these aspects are case-sensitive, there is no way that the results can be generalized. These consequences must be considered, and experimental procedures must be optimized [45]. 

#### 3.2.3. Determination of Cytokines (IL-6 and TNF-α) in RAW 264.7 and THP-1 Cell Line

TNF-α is considered a master regulator of inflammatory cytokine cascade in immune responses playing a protective role against viral, protozoan, or bacterial infection [46,47,48,49] and other inflammatory diseases [50,51,52,53]. THP1 cells—which are considered a human monocyte and macrophage model in vitro for studying immune responses—derived into macrophages increase TNF-α production in the presence of *F. chilensis* polysaccharides, revealing that these molecules can modulate the immune system and contribute to inflammatory responses [54]. The findings demonstrate that FcPs elicited elevated production of IL-6 and TNF-α in the RAW 264.7 macrophage cell line, two proinflammatory cytokines relevant to innate immune response. Thus, it implies that FcPs may exhibit a stimulatory impact on the immune system by triggering macrophages to secrete these cytokines. The findings indicate that the concentration of polysaccharides plays a crucial role in the production of IL-6 and TNF-α. Specifically, the synthesis of IL-6 increases from 4000 to 7500 pg mL^−1^ when the polysaccharide concentration rises from 5 to 25 μg mL^−1^. However, this effect saturates above the mentioned concentration level. Therefore, there exists a threshold of stimulation for IL-6, and higher polysaccharide concentrations do not have an extra impact. When examining TNF-α, it becomes evident that the production of this cytokine significantly rises from 1.0 mg mL^−1^, with a maximum increase of 7000 pg mL^−1^ seen at 2.0 mg mL^−1^ concentration. The findings suggest that TNF-α is stimulated with a delay, and lower concentrations of polysaccharides do not have a meaningful impact. The precise process by which β-glucans suppress inflammatory cytokines and induced anti-inflammatory cytokines is intricate and not completely comprehended [55]. The observed outcomes may be attributed to the interaction of FcPs with various receptors or signaling pathways in macrophages, which exhibit distinct sensitivities and affinities [56]. Fungal polysaccharides, for instance, can activate Toll-like receptors (TLRs), Dectin-1, Complement receptor 3 (CR3), Toll-like receptors (TLRs), Lactosylceramides, and scavenger receptors [56,57]. These receptors can trigger the production of IL-6 and TNF-α, but can also lower their expression [58]. FcPs may have a double-edged effect on macrophages, as they appear to stimulate cytokine production at specific concentrations while inhibiting it at others.

#### 3.2.4. HCT116 Cell Cycle Analysis Using Flow Cytometry

The study was conducted using HCT116 cells treated with FcPs, and it showed a significant increase in the Sub G0/G1 phase. This increase indicates that the FcPs induced an apoptotic population in a dose-dependent manner. It was also observed that this increase in apoptosis was accompanied by a decrease in the G2/M phase [59]. Polysaccharides have been extensively studied for their anti-tumor properties, including mechanisms such as cell cycle arrest, anti-angiogenesis, and apoptosis, which directly contribute to tumor suppression [60]. Apoptosis, which is a crucial process in cancer development and progression, has been associated with the effects of various polysaccharides on different cancer cell lines [61,62]. It has been demonstrated that polysaccharides can induce apoptosis in cancer cells through various pathways, including the mitochondrial apoptotic pathway and modulation of apoptotic proteins [63,64,65]. The findings of the study support existing research indicating the ability of FcPs to induce apoptosis in cancer cells. The modulation of apoptotic pathways by polysaccharides, as observed in the study on FcPs, further supports the therapeutic potential of these natural compounds in cancer treatment.

#### 3.2.5. Proteomic Analysis in HGF-1 Cells

The use of polysaccharides from *F. chilensis* as therapy in diabetic cardiomyopathy could contribute to improving this heart disease thanks to its suppressive effect on both the expression of type I collagen and NADH: ubiquinone oxidoreductase, as well as its activating effect on the expression of Phospholipase c.

Myocardial fibrosis caused by metabolic abnormalities is the initial change associated with diabetic cardiomyopathy [66]. Fibroblasts are thought to be responsible for the production and degradation of the extracellular matrix in the interstitial tissue, and they play an important role in retaining cardiac muscle structure [67]. Fibrillar collagen is the main component of the extracellular matrix in the heart, where type I collagen accounts for 80% [68]. The proteomic analysis showed that the treatment of HGF-1 cells with polysaccharides from *F. chilensis* modifies the expression of the type I collagen, causing its decrease, which suggests that it could contribute to improving diabetic cardiomyopathy. Moreover, Phospholipase C (PLC) activity is known to influence cardiac function. Different experimental models of chronic diabetes that examined the status of PLC beta3 in the cardiac cell plasma membrane (sarcolemma) suggested that a decrease in PLC beta3 protein abundance may contribute to the cardiac dysfunction seen during diabetes [69]. Treatment with polysaccharides from *F. chilensis* leads to the overexpression of PLC beta3, which could reverse abnormal heart function in diabetes. All of these findings justify the traditional use of these mushrooms in the treatment of heart disease.

### 3.3. STRING Analysis of Protein Networks

The overexpression of Glutathione-S-transferase mu 2 and Superoxide dismutase 2 in normal cells after treatment with FcPs could explain their antioxidant and cellular detoxifying effects.

Drug-metabolizing enzymes (DMEs), such as glutathione S-transferases (GSTs), play an important role in scavenging the waste products of lipid and oxidative metabolism [70]. Glutathione-S-transferase mu 2 (GSTM2), as a member of DMEs, promotes cellular anti-oxidation and detoxification. Moreover, GSTM2 was found highly upregulated in high-fat diet-fed mice, and the loss-of-function GSTM2 mouse model demonstrated that GSTM2 protected mice from excess fat accumulation [71]. Our study showed overexpression of GSTM2 when normal HGF-1 cells were treated with polysaccharides from *F. chilensis*, which could imply a potential therapeutic role as an antioxidant and cellular detoxifier, as well as in excess fat accumulation. Furthermore, concomitant overexpression of superoxide dismutase 2 (SOD2), a key component of the antioxidant defense system against mitochondrial superoxide radicals [72], could enhance the antioxidant effects of these naturally occurring polysaccharides.

The transcription factor nuclear factor-κB (NF-κB) modulates immune cell functions and alters the gene expression profile of different cytokines in response to various pathogenic stimuli [73]. The activation and functions of NF-κB have been known to be modulated by a family of proteins known as NF-κB inhibitors (IκBs) [74]. The primary mechanism for canonical NF-κB activation is the inducible degradation of IκBα triggered through its site-specific phosphorylation by a multi-subunit IκB kinase (IKK) complex. IKK comprises two catalytic subunits, IKKα and IKKβ [7]. Our study shows that the treatment of normal cells with polysaccharides from *F. chilensis* increases the expression of the IKBKB (synonym of IKKβ) by more than six times, which could contribute to an increase in the activation level of NF-κB and thus to increased levels of inflammatory cytokines, such as those observed in polysaccharide-treated macrophages in the present study.

## 4. Materials and Methods

### 4.1. Materials and Reagents

Standard monosaccharides (D-galactose, D-glucose, D-mannose, L-rhamnose, X-fructose, Y-apiose, D-xylose, and myo-inositol) were purchased from Sigma Aldrich, St. Louis, MO, USA. The following reagents were of analytical grade: 1,1-diphenyl-2-picry-hydrazyl (DPPH*), and ABTS (2,20-Azinobis- (3-ethylbenzthiazoline-6-sulphonate). All other chemicals (pyridine, hexane, and methanol/3 M HCl solution) were analytical grade and were purchased from either Merck (Darmstadt, Germany) or Sigma-Aldrich (St. Louis, MO, USA). All water was obtained from a Milli-Q system (Millipore, Billerica, MA, USA). Other materials were purchased from local suppliers.

### 4.2. Collection, Authentication of Basidiocarps and Mycelial Culture

The carpophore of *Fomitiporia chilensis* was collected from a *Cryptocarya alba* (Peumo) specimen in the Coyanmahuida Park, in Commune of Florida, province of Concepción (Biobío Region, Chile). The identity of the basidiome was confirmed based on macro- and micro-morphological features as described in [1] by CR (co-author of the aforementioned publication). A representative voucher specimen (Accession CONC-F 1926) was deposited at the herbarium in the same department of the University of Concepción. The mycelial culture was maintained using YMG agar medium (0.5% *w*/*v* yeast extract (B.D. Biosciences, San José, CA, USA), 1% *w*/*v* malt extract (B.D. Biosciences, San José, CA, USA), 1.5% *w*/*v* glucose, and 2% *w*/*v* agar) by culturing at 20 ± 2 °C. Later, mycelium fragments were transferred to liquid YMG medium (1% *w*/*v* glucose, 1% *w*/*v* malt extract, 0.4% *w*/*v* yeast extract) at pH 5.8 and incubated at 20 °C for 1 to 2 months under constant agitation (120 rpm). Finally, the mycelium was lyophilized (Lyophilizer Cryodos, Telstar, Terrasa, Spain).

### 4.3. Extraction of Polysaccharides

Polysaccharide extraction was conducted by Parages et al. (2012) [75]. In total, 10 g of lyophilised *F. chilensis* biomass was suspended in 400 mL of 80% (*v*/*v*) EtOH (Sigma-Aldrich, St. Louis, MO, USA) and agitated at room temperature for 16 h. After this pre-treatment, the solution was centrifuged at 4000 rpm for 10 min. The supernatant of the solution was removed before resuspension of the pellet in 400 mL of 80% EtOH. To extract lipophilic pigments, this suspension process was repeated multiple times until we obtained a colorless supernatant. The final pellet was re-suspended in 300 mL of distilled water and stirred at 100 °C. Once boiling temperature was reached, the heating process was terminated, and the solution was agitated for more than 30 min. This procedure was performed thrice. The resulting mixture was subsequently centrifuged (3000 rpm) for 15 min. The presence of phenols was confirmed using the Folin-Ciocalteu reagent. Thereafter, polyvinylpyrrolidone precipitation and centrifugation (at 3000 rpm for 5 min) were used to eliminate the phenols. The extraction of polysaccharides from the supernatant involved selective precipitation using 2% O-N-cetyl pyridinium bromide (Cetavlon, Sigma Chemical Co., St. Louis, MO, USA), as described in Casas-Arrojo et al., 2021 [76]. The acid polysaccharides precipitated were purified with 4 M NaCl, then flocculated with 96% (*v*/*v*) EtOH (Sigma-Aldrich, St. Louis, MO, USA) and centrifuged at 4500 rpm for 10 min. The recovered pellet was dialyzed overnight at low osmotic pressure (0.5 M NaCl) and undertook agitation at 4 °C on a dialysis membrane (Dialysis Tube Cellulose, Sigma-Aldrich, St. Louis, MO, USA). Following dialysis, the entire membrane content was retrieved and rinsed with 96% EtOH. The mixture was maintained at a temperature of 4 °C until the flocculation of the polysaccharides occurred. Subsequently, a final centrifugation of 2 min at room temperature was conducted at 3000 rpm to retrieve the polysaccharides. Next, the polysaccharides were frozen at a temperature of −80 °C and then subjected to lyophilization with the Cryodos Lyophilizer from Telstar, Terrasa, Spain. 

### 4.4. Total Carbon (C), Hydrogen (H), Nitrogen (N), and Sulfur (S), and Protein Content

The total carbon (C), nitrogen (N), and sulfur (S) contents were analyzed in the dry biomass of *F. chilensis* and extracted polysaccharides, utilizing the total combustion method according to Abdala-Díaz et al. [77]. This technique is based on the complete and instantaneous oxidation of the sample by pure combustion with controlled oxygen at a temperature of 1050 °C (C, H, N, S), followed by pyrolysis at 1300 °C (O) for decomposition of O as CO and oxidation to CO_2_. The resulting combustion products, CO_2_, H_2_O, SO_2_, and N_2_, were subsequently quantified using the selective I.R. absorption detector (C, H, S), as well as the TCD (N) differential thermal conductivity sensors of the LECO TruSpec Micro CHNSO-elemental analyzer (St. Joseph, MI, USA). The result obtained for each element (C, H, N, S) was expressed in percentages (%) concerning the weight of the sample. Once the samples were analyzed with the Perkin-Elmer 2400 Mar Biotechnol analyzer (Perkin-Elmer, Waltham, MA, USA). The protein content was calculated by multiplying the N% by the N-Prot factor. Since for *F. chilensis* the factor was not determined, the global mean N-Prot factor of 6.25 for casein of milk was used.

### 4.5. Fourier Transform Infrared Spectroscopy (FT-IR)

Self-supporting pressed disks (16 mm in diameter) composed of a mixture of the polysaccharides and KBr (1% *w*/*w*) were made with a hydrostatic press at a force of 15.0 tcm^−2^ for 2 min. The FT-IR spectra of the polysaccharides fraction of *F. chilensis* were acquired using a Thermo Nicolet Avatar 360 IR spectrophotometer (Thermo Electron Inc., Waltham, MA, USA) having a resolution of 4 cm^−1^ with a DTGS detector and using an OMNIC 7.2 software (with a bandwidth of 50 cm^−1^ and an enhancement factor of 2.6). The region of 400–4000 cm^−1^ was utilized for the measurements. Baseline adjustment was performed using the Thermo Nicolet OMNIC software to flatten the baseline of each spectrum. Furthermore, the OMNIC correlation algorithm was used to compare sample spectra with those of the spectral library (Thermo Fischer Scientific, Waltham, MA, USA).

### 4.6. Gas Chromatography-Mass Spectrometry (GC-MS)

#### 4.6.1. Derivatization of Polysaccharides

Polysaccharides extracted from *F. chilensis* (2 mg), and monosaccharide standards underwent the same procedure. Initially, 100 µL of the standard stock solution containing 1 mg mL^−1^ of each monosaccharide was dried under nitrogen gas flow. The polysaccharide samples, along with a mixture containing standard monosaccharides from the I.S., were treated in a 2 mL HCl 3 M solution in MeOH at 80 °C for 24 h. The saccharides were washed using methanol and dried under a flow of nitrogen gas. Following this, the trimethylsilyl reaction was carried out using 200 µL of Tri-Sil HTP (Thermo Scientific). Each vial containing the sample was heated to 80 °C for 1 h. After the sample was derivatized, it was cooled to room temperature and dried using nitrogen. The dry residue was then extracted with 2 mL of hexane and centrifuged. Finally, the solution of silylated monosaccharides in hexane was concentrated, reconstituted (200 µL), and filtered before being transferred to a GC-MS autosampler vial. The sample was prepared and analyzed in triplicate. 

#### 4.6.2. Gas Chromatography/Mass Spectrometry (GC-MS) Analysis

GC-MS analyses were carried out using a gas chromatography Trace GC (Thermo Scientific), autosampler Tri Plus, and DSQ mass spectrometer quadrupole (Thermo Scientific). The column was ZB-5 Zebron, Phenomenex (5% Phenyl, 95% Dimethylpolysiloxane, Torrance, CA, USA), of dimensions 30 m × 0.25 mm i.d. × 0.25 µm. The column temperature was programmed from 80 °C (held 2 min) and 5 °C min-1 to a final temperature of 230 °C. The carrier gas was helium (flow 1.2 mL/min). The injection volume was 1 µL in a splitless mode at 250 °C. The source and M.S. transfer line temperature were at 230 °C. A Select Ion Monitoring Program (SIM) in electron ionization mode was set at 70 electron volts (eV) in the mass spectrometer. The TMS-derivatives were identified using characteristic retention times and a mass spectrum compared to those of authentic standards. The compounds were identified by comparing the mass spectra with those in the library of the National Institute of Standards and Technology (NIST, 2014). 

### 4.7. Antioxidant Activity (ABTS) Free-Radical Method in Biomass from F. chilensis

The ability of the polysaccharides to scavenge free radicals was evaluated using an ABTS assay according to Re et al. (1999) [78], with a few modifications. Antioxidant capacity as 2,2′-azino-bis (3-ethylbenzothiazoline-6-sulfonic acid (ABTS)) Assay Scaveng-ing of Free Radical in polysaccharides. An aqueous solution containing 7 mM ABTS (Sigma-Aldrich, St. Louis, MO, USA) was mixed with 2.45 mM potassium persulfate (Sigma-Aldrich, St. Louis, MO, USA) for 16 h in the dark at room temperature. After incubation, the well-mixed solution was diluted to an absorbance of 0.7 at 727 nm with deionized water. The final concentrations of polysaccharide solution were: 25, 50, 75, 100, 150, 200, 300, 400, and 500 μg mL^−1^. A total of 50 μL of these samples were mixed with 950 μL of ABTS solution. The resulting mixture was measured spectrophotometrically at 727 nm (Micro Plate Reader 2001, Whittaker Bioproducts, Walkersville, MD, USA). ABTS radical scavenging capacity was calculated according to the following Equation:AA% = [(A_0_ − A_1_)/A_0_] × 100 
where A_0_ is the absorbance of the ABTS radical in phosphate buffer at time 0 and A_1_ is the absorbance of the ABTS radical solution mixed with the sample after 8 min. All determinations were performed in triplicate (*n* = 3). 

### 4.8. In Vitro Immunomodulatory Activity Assay

#### 4.8.1. Cell Culture

In this study, five cell lines were utilized, including the colon cancer cell line (HCT-116, ATCC, Manassas, VA, USA), the human leukemia cell line (U-937, Manassas, VA, USA), the human monocytes cell line (THP-1, Manassas, VA, USA), the murine macrophages cell line (RAW 264.7, ATCC, Manassas, VA, USA), and the human gingival fibroblasts cell line (HGF-1, primary culture). RAW 264.7, HCT-116, and HGF-1 cells were routinely cultured in Dulbecco’s modified Eagle’s medium (DMEM) (Biowest, Nuaillé, France) that was supplemented with 10% fetal bovine serum (FBS) (Biowest, Nuaillé, France), a 1% penicillin-streptomycin solution 100 UI/mL (Biowest, Nuaillé, France), and 0.5% of amphotericin B (Biowest, Nuaillé, France). Meanwhile, while U-937 cells were cultured in RPMI-1640 medium supplemented with 10% fetal bovine serum, 1% penicillin-streptomycin solution 100 UImL^−1^, and 0.5% of amphotericin B and THP-1 cells were cultured in RPMI-1640 medium (Biowest, Nuaillé, France) which was supplemented with 0.05 mM 2-mercaptoethanol (Sigma-Aldrich, St. Louis, MO, USA), 10% fetal bovine serum, 1% penicillin-streptomycin solution 100 UI mL^−1^, and 0.25 μg mL^−1^ of amphotericin B. Cells were maintained sub-confluent at 37 °C in a humid atmosphere containing 5% CO_2_. Cultured cells were collected using gentle scraping when confluence reached 75% in HTC-116 and HGF-1, as they are adherent cells. Scrapping of U-937 cells was not performed because these cells are in suspension. These cells were collected using centrifugation at 1500 rpm for 5 min.

#### 4.8.2. Cell Viability Assay of Lines Raw 264.7, U-937, HTC-116, and HGF-1

The Raw 264.7, U-937, HTC-116, and HGF-1 cells were incubated with different concentrations of FcPs (0–100 μg mL^−1^). The experiment was conducted individually for each cell line in a 96-well microplate for 72 h (37 °C, 5% CO_2_ in a humid atmosphere). As a positive control, the same cell lines were used without treatment. An MTT assay (3-(4,5-dimethylthiazol-2-yl)-2,5-diphenyltetrazolium bromide) was utilized to estimate the proliferation of these cell lines (Abdala Díaz et al., 2011) [79]. A volume of 10 μL of the MTT solution (5 mg mL^−1^ in phosphate-buffered saline) was added to each well. Incubation of the plates was then carried out at 37 °C for 4 h. Formazan was dissolved with the addition of acid-isopropanol (150 μL of 0.04 N HCl-2- propanol) and measured using spectrophotometry at 550 nm (Microplate reader, Biotek, Synergy HT, VT, USA). The cell viability was expressed as the mean percentage of viable cells compared with untreated cells. Four replicates were performed for each concentration tested across three independent experiments.

The selectivity index determines the cytotoxic selectivity of the compounds tested and is calculated by the ratio between the IC_50_ as indicated in the following formula: SI = (IC_50_ of health cell)/(IC_50_ of cancer cell) 

According to Indrayanto et al. [39], a selective compound presents a SI over 3.

#### 4.8.3. Cell Cycle Analysis by Flow Cytometry (HCT-116 Human Cancer Cell)

HCT-116 cells were seeded in six-well plates and incubated at 37 °C in a humidified chamber with 5% CO_2_ until subconfluence. Subsequently, the corresponding treatment was added to the fresh complete culture medium, and cells were incubated at the same conditions for 16h. Following the treatment, cells were harvested and centrifuged. Pellets were washed with PBS supplemented with 1% FBS and HEPES 10 mM and then fixed using 70% EtOH, for 1 h at −20 °C. Finally, the cells were centrifuged and washed twice with PBS supplemented with 1% FBS and HEPES 10 mM. They were then suspended in a propidium iodide staining solution containing 40 μg mL^−1^ propidium iodide, 0.1 mg mL^−1^ RNase-A, and EDTA 2 mM in PBS supplemented with 1% FBS and HEPES 10 mM. Subsequently, they were incubated for 30 min at 37 °C in a light-protected chamber. The samples were measured using a FACS VERSETM flow cytometer (B.D. Biosciences), and the results were analyzed using the Kaluza analysis software version 2.1 (Beckman Coulter, Brea, CA, USA).

#### 4.8.4. Determination of Cytokines with RAW 264.7 Cell Line

The RAW 264.7 cells were cultured in 96-well microplates (5 × 10^4^ cells well^−1^) using Dulbecco’s modified Eagle’s medium (DMEM) at a temperature of 37 °C, 5% CO_2_, and a humid atmosphere, and in the presence of different concentrations (0–100 μg mL^−1^) of polysaccharides in a total volume of 100 µL. Conditioned media from the untreated and treated RAW cells over 24 h were collected. Interleukin-6 (IL-6) present in these conditioned media was quantified by using a mouse IL-6 ELISA Ready-SET-Go Kit (Affymetrix, EBioscience, Madrid, Spain), following the supplier’s instructions.

#### 4.8.5. Determination of Cytokines with THP-1 Cell Line

Human THP-1 cells were seeded in 96-well microplates (5 × 10^4^ cells per well^−1^) and differentiated into macrophages after a 10 ng mL^−1^ PMA stimulation for 48 h. Subsequently, wells were washed with fresh medium and exposed to different concentrations of the polysaccharide for 24 h. Human TNF-α present in the conditioned media was quantified by using a Human TNF alpha uncoated ELISA kit (Invitrogen, Thermo Fisher Scientific, Madrid, Spain).

### 4.9. Proteomics Analysis (UHPLC-HRMS Analysis for Differential Protein Expression Detection in Treated HGF-1 Cells)

An analysis using ultra-high-performance liquid chromatography high-resolution mass spectrometry (UHPLC-HRMS) was conducted on HGF-1 cells to investigate the effects of polysaccharides obtained from *F. chilensis* on protein expression levels in non-cancerous cells.

#### 4.9.1. Cell Treatment and Protein Extraction 

HGF-1 cells were cultured in DMEM without fetal bovine serum and treated with 1.0 mg mL^−1^ of polysaccharides for 24 h. Afterward, cells were rinsed with ice-cold phosphate-buffered saline and solubilized in 200 μL of RIPA buffer. Then, cell extracts were sonicated, centrifuged at 14,000× *g*, and purified using a modified trichloroacetic acid protein precipitation procedure (Clean-Up Kit; G.E. Healthcare, München, Germany). Afterward, gel-assisted proteolysis was carried out. Briefly, the protein solution was entrapped in a polyacrylamide gel matrix before reduction with dithiothreitol and cysteine residue alkylation with iodoacetamide. Then, proteins were digested with trypsin (Promega, Madison, WI, USA), and peptides were extracted from the gel and purified using a C18 ZipTip (Merck Millipore, Darmstadt, Germany) using the manufacturer’s protocol.

#### 4.9.2. Liquid Chromatography High-Resolution Mass Spectrometry

The Ultra-High-Performance Liquid Chromatography and the High-Resolution Mass Spectrometry analysis were carried out as described elsewhere (Ojeda et al., 2021) [80]. Briefly, samples were injected into an Easy nLC 1200 UHPLC system coupled to a Q Exactive™ HF-X Hybrid Quadrupole-Orbitrap mass spectrometer (Thermo Fisher Scientific, Waltham, MA, USA). Peptides extracted from the samples were then loaded automatically into a trap column and eluted onto a 50 cm analytical column with a 180 min gradient at a flow rate of 300 nL min^−1^. Data acquisition was performed in electrospray ionization positive mode using a data-dependent acquisition method to obtain the corresponding MS/MS spectra.

#### 4.9.3. Data Analysis

The acquired raw data were analyzed using the Proteome DiscovererTM 2.3 platform (Thermo Fisher Scientific, Waltham, MA, USA). For the identification of the MS2 spectra, Sequest HT^®^ was utilized as the search engine, and the Swiss-Prot part of UniProt for Homo sapiens was used as the database [81]. Protein assignments were validated using the Percolator^®^ algorithm by imposing a strict cut-off of 1% false discovery rate (FDR) [82].

Label-free quantitation was implemented using the Minora feature of Proteome DiscovererTM 2.3. Abundances were based on precursor intensities. Normalization was performed based on total peptide amount. Protein abundance ratios were calculated based on unique peptides as the median of all possible pairwise ratios calculated between replicates of all connected peptides. Abundance ratio p-values were calculated using ANOVA based on background population of peptides and proteins. Proteins with ANOVA *p* < 0.05 and Log2 fold change > 0.5 were considered to be significantly deregulated.

The Search Tool for the Retrieval of Interacting Genes/Proteins (STRING) (https://www.string-db.org, accessed on 14 November 2022) was used to perform a protein-protein interaction network functional enrichment bioinformatics analysis. Functional pathways were analyzed using the Kyoto Encyclopedia of Genes and Genomes (KEGG) database (http://www.genome.jp/kegg/pathway.html, accessed on 14 November 2022).

### 4.10. Statistical Analysis 

Means ± standard deviation (S.D.) was used to express all values. Differences among treatments with polysaccharides were evaluated using a one-way ANOVA. When significant differences were found, a post hoc Tukey HSD test was conducted to compare the means among treatments. Differences were considered statistically significant at *p* < 0.05. All statistical analyses were performed using the Statistica 7.0 software. Figures were generated using SigmaPlot version 12.0, 2015 (Systat Software Inc., Richmond, CA, USA).

## 5. Conclusions

This study conducted on the polysaccharides derived from the fungus *Fomitiporia chilensis* has yielded some fascinating results concerning its biological activity and potential medicinal benefits. The acidic polysaccharides extracted from *F. chilensis* were analyzed for their antioxidant capacity, immunomodulatory activity, anti-tumor, and cytotoxicity activity. Additionally, a cell cycle analysis, chemical analysis, and proteomic analysis provided insights into the molecular mechanisms underlying the antioxidant and cellular detoxifying effects of these polysaccharides. In conclusion, this study suggests that the acidic polysaccharides found in *F. chilensis* may have tremendous potential as a candidate for cancer therapy and immune system modulation. Further research may help to elucidate the precise mechanisms of action and contribute to the advancement of human health.

## Figures and Tables

**Figure 1 molecules-29-03628-f001:**
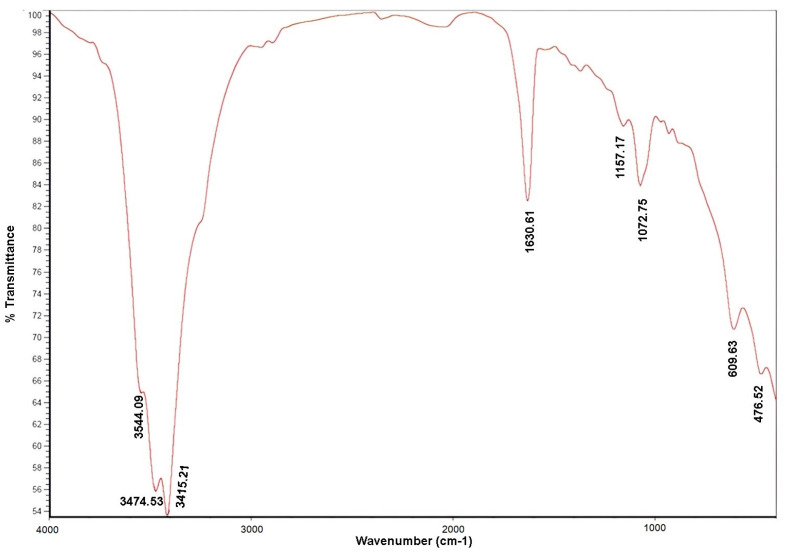
FT-IR analysis of the polysaccharide’s fraction obtained from *Fomitiporia chilensis*.

**Figure 2 molecules-29-03628-f002:**
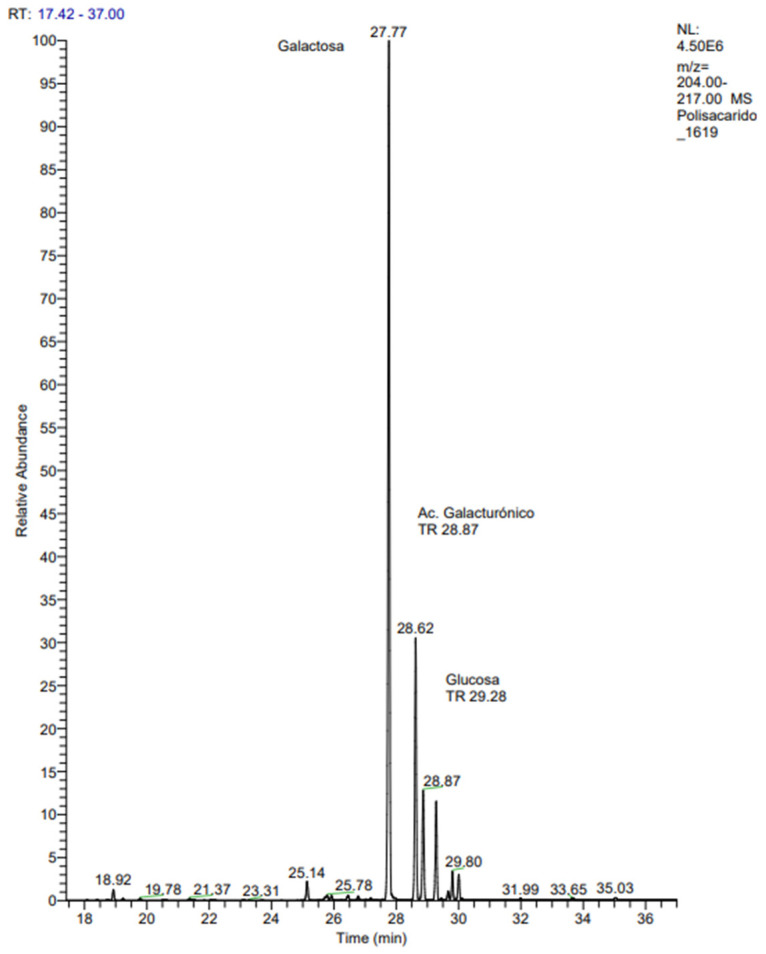
The GC-MS sugar analysis method was employed to examine the monosaccharide components of *F. chilensis* polysaccharides (FcPs). The main peak at a retention time of 27.77 min represents Gal, the peak at a retention time of 28.62 min represents GalA, and the peak at a retention time of 28.87 and 29.28 represents glucose (Glc) isomer α and β when compared with the retention time of the standard monosaccharide.

**Figure 3 molecules-29-03628-f003:**
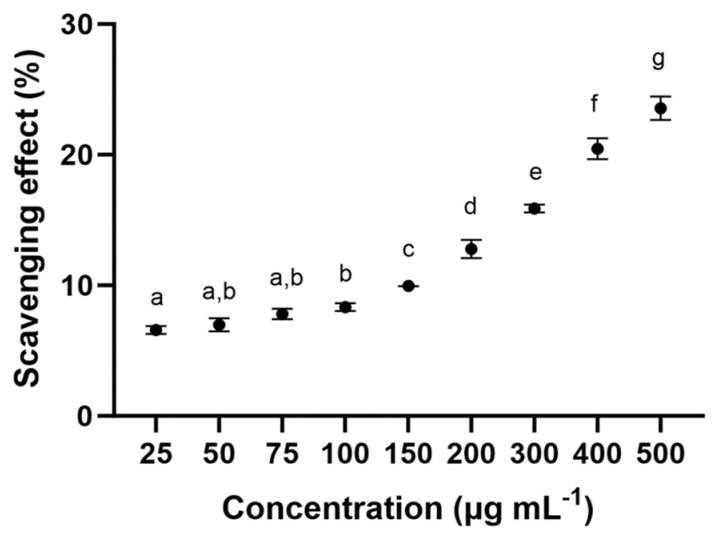
Scavenging effects % of the sample on ABTS radical. The data are the mean of three replicate measurements ± standard error. Similar letters indicate no significant differences (Tukey, *p* < 0.05) between the different concentrations.

**Figure 4 molecules-29-03628-f004:**
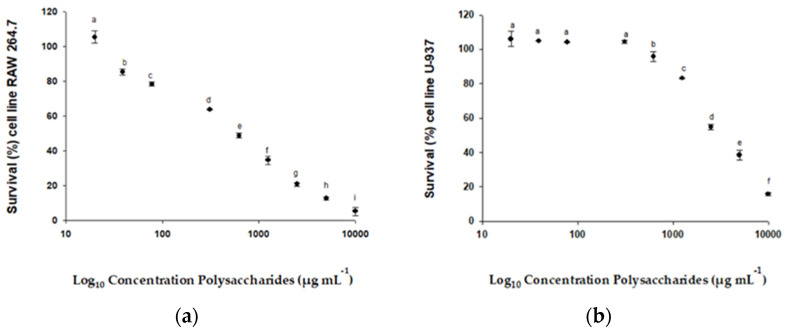
% Survival of different cell lines exposed to different concentrations of polysaccharides from *F. chilensis*: (**a**) % Survival of cell RAW 264.7; (**b**) % Survival of cell U-937; (**c**) % Survival of cell HTC-116; (**d**) % Survival of cell HGF-1. Similar letters indicate no significant differences (Tukey, *p* ≤ 0.05) between the different concentrations.

**Figure 5 molecules-29-03628-f005:**
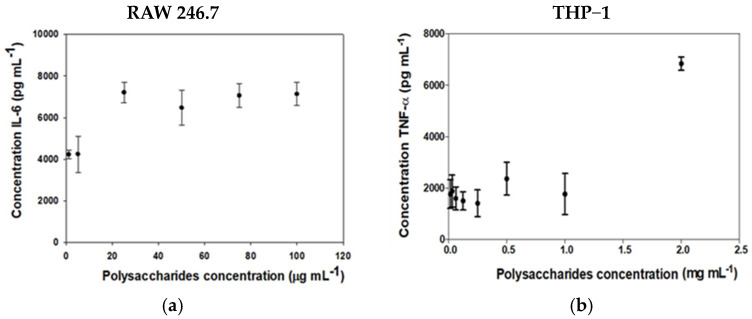
(**a**) IL-6 production was measured in RAW 246.7 macrophages after 24 h of exposure to various FcPs concentrations; (**b**) TNF-α concentration was measured in THP1-derived macrophages similarly treated with FcPs. To determine cytokine quantities in conditioned media, ELISA assays were conducted. The resulting data show the mean ± S.D. of three independent experiments.

**Figure 6 molecules-29-03628-f006:**
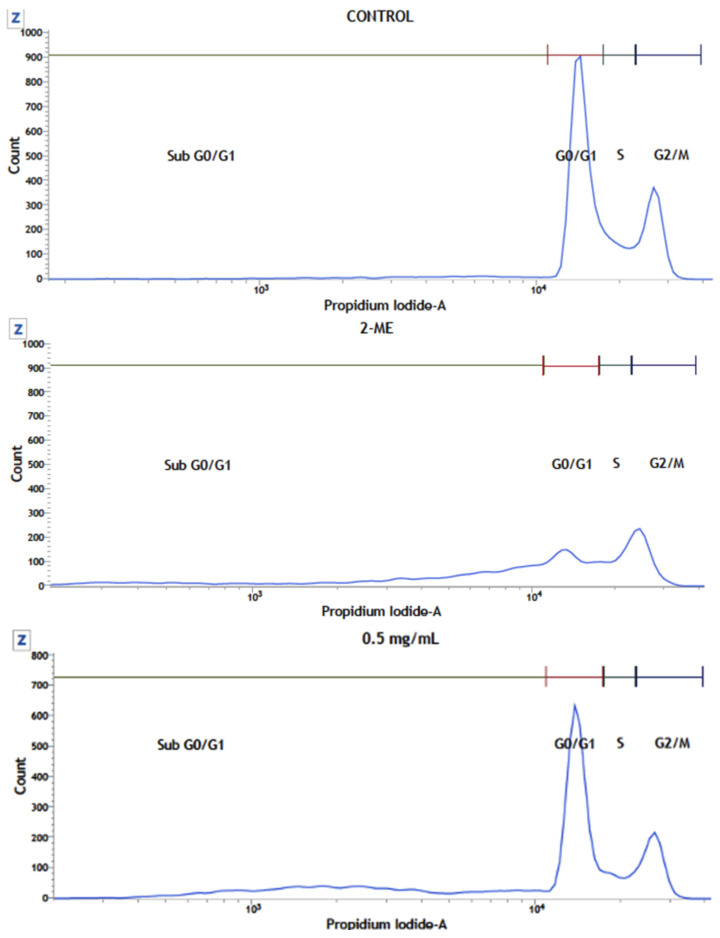
Effect of FcPs on the HCT166 Cell Cycle. (**A**) Representative cell cycle histograms for HCT166 control conditions or treated with the polysaccharide at the pointed concentrations, measured after 16 h using propidium iodide staining and flow cytometry. (**B**) The quantitative analysis for the four cell cycle subpopulations in control and treated conditions. Data are means ± S.D. of at least three independent experiments (* *p* < 0.05, ** *p* < 0.01).

**Figure 7 molecules-29-03628-f007:**
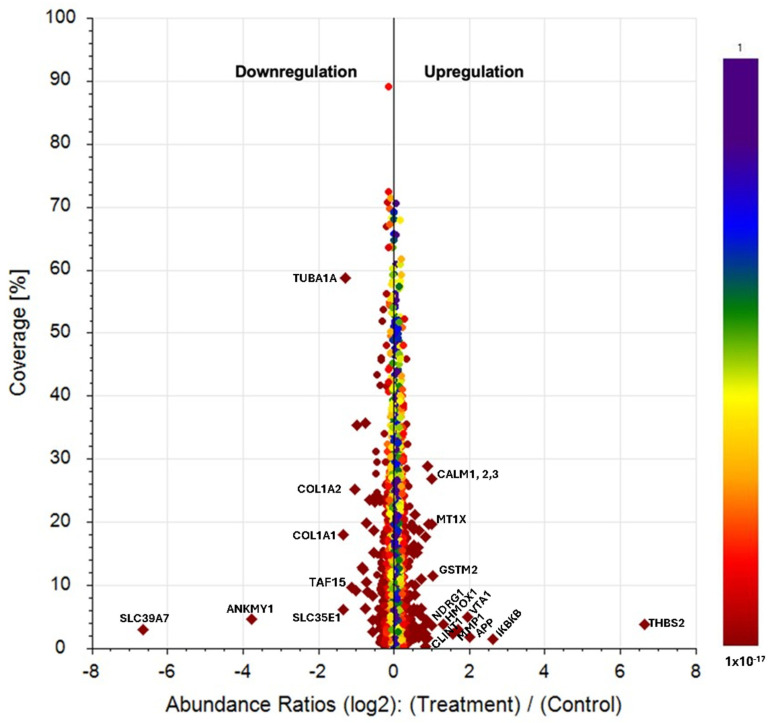
Scatter plots showing the percentage of protein sequence coverage vs. their Log2 fold change abundance value in HGF cells after 24 h with FcPs. Proteins were ranked according to their *p*-value from red to blue color. Proteins with *p*-value < 0.05 and Log2 fold change < −0.5 or >0.5 are drawn in diamonds (only gene symbols with fold change < 0.5 or >2 are shown).

**Figure 8 molecules-29-03628-f008:**
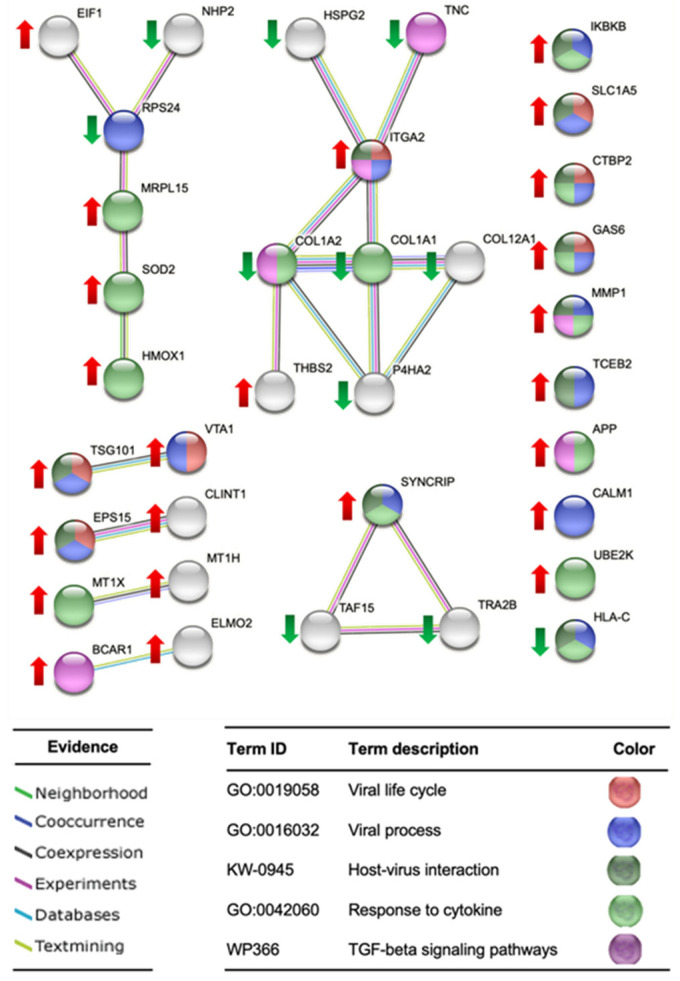
Protein-protein interaction networks functional enrichment analysis of significantly deregulated proteins in human gingival fibroblasts treated with polysaccharides from the fungi *F. chilensis*. This figure presents a representative partial list of the significantly enriched Gene Ontology (G.O.), WikiPathways (W.P.), and UniProt annotated keywords (K.W.) pathways associated with the deregulated proteins using STRING (Search Tool for the Retrieval of Interacting Genes/Proteins). Network nodes are proteins, and edges represent the predicted functional associations. The proteins involved in the different processes are listed in Table 3 and colored in Figure 9 according to their function. The edges are drawn according to the type of evidence. Overexpressed (abundance ratio Log2 > 0.5) and under-expressed (abundance ratio Log2 < 0.5) proteins have been marked with a red up arrow and a green down arrow, respectively.

**Figure 9 molecules-29-03628-f009:**
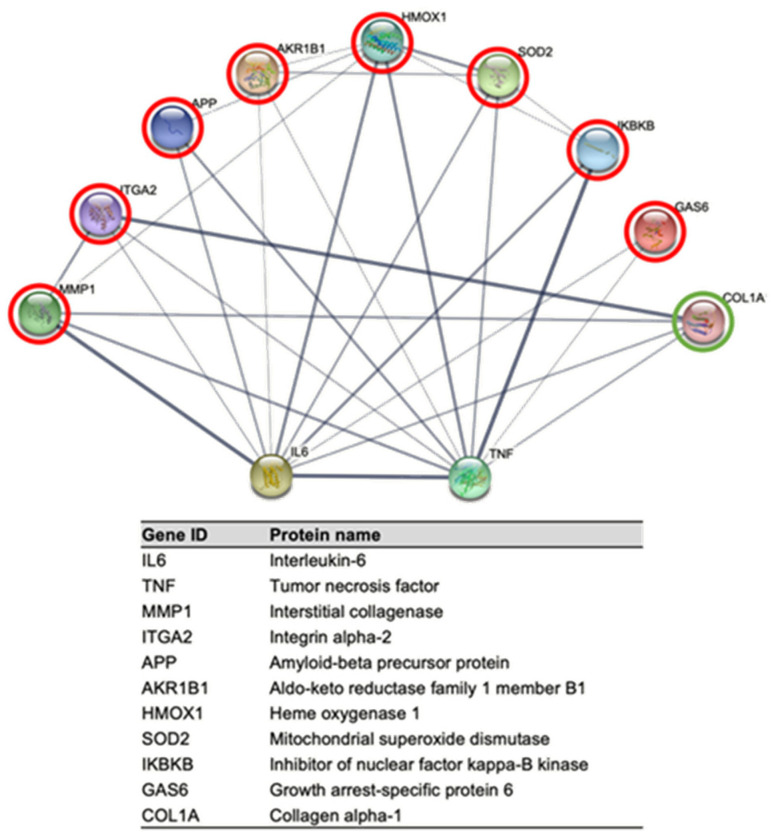
Known functional interactions among the significantly deregulated proteins and the key immunological markers IL-6 and TNF-α. Line thickness indicates the strength of data support. Several deregulated proteins functionally interact with IL-6 and TNF-α. Overexpressed and under-expressed proteins are circled in red and green, respectively.

**Table 1 molecules-29-03628-t001:** Total Carbon (C), Nitrogen (N), Sulfur (S), and protein content.

	Extracted Polysaccharides (%)
Carbon	40.03
Hydrogen	6.08
Nitrogen	2.14
Sulfur	0.97
Protein (dry weight, D.W., %)	18.24

**Table 2 molecules-29-03628-t002:** Content of monosaccharides in FcPs.

Header	Retention Time (min)	Monosaccharide	Peak Area	% Mass
1	27.77	Galactose (Gal)	21.099.271	53.02
2	28.62	Galacturonic acid (GalA)	6.198.261	16.02
3	28.87	Glucose (Glc α)	2.772.784	6.83
4	29.28	Glucose (Glc β)	2.406.072	6.27

**Table 3 molecules-29-03628-t003:** Proteins significantly deregulated in HGF-1 cells after FcPs treatment.

Gene Symbol	Description	Sum PEP Score ^(1)^	Abundance Ratio: Treatment/Control	Abundance Ratio *p*-Value
THBS2	Thrombospondin-2	12.61	^(2)^	1.00 × 10^−17^
**IKBKB**	Inhibitor of nuclear factor kappa-B kinase subunit beta	2.88	6.16	1.00 × 10^−17^
APP	Amyloid-beta A4 protein	2.92	4.08	1.00 × 10^−17^
VTA1	Vacuolar protein sorting-associated protein VTA1 homolog	3.55	3.92	1.00 × 10^−17^
MMP1	Interstitial collagenase	3.22	3.26	1.00 × 10^−17^
CLINT1	Clathrin interactor 1	3.77	2.96	1.00 × 10^−17^
HMOX1	heme oxygenase 1	2.88	2.51	1.00 × 10^−17^
**GSTM2**	Glutathione S-transferase Mu 2	5.85	2.07	3.96 × 10^−8^
NDRG1	Protein NDRG1	6.02	2.03	3.17 × 10^−11^
MT1X	metallothionein-1X	6.31	2.03	2.29 × 10^−13^
CALM1, 2, 3	Calmodulin	21.48	2.03	2.76 × 10^−13^
MT1H	MX10tallothionX10in-1H	3.84	1.92	1.47 × 10^−8^
**PLCB3**	1-phosphatidylinositol 4,5-bisphosphate phosphodiesterase beta-3	2.59	1.88	1.37 × 10^−7^
AKR1B1	aldose reductase	25.63	1.85	2.75 × 10^−10^
MRPL15	39S ribosomal protein L15, mitochondrial	3.39	1.85	2.67 × 10^−8^
LOXL2	Lysyl oxidase homolog 2	3.66	1.79	2.39 × 10^−6^
DSP	Desmoplakin	2.85	1.79	3.61 × 10^−9^
AKR1C1	Aldo-keto reductase family 1 member C1	20.00	1.79	3.10 × 10^−9^
SKIV2L	Helicase SKI2W	2.87	1.74	2.86 × 10^−5^
ELMO2	Engulfment and cell motility protein 2	5.25	1.71	2.74 × 10^−5^
HPCA	Neuron-specific calcium-binding protein hippocalcin	6.12	1.67	3.25 × 10^−6^
**SOD2**	Superoxide dismutase [Mn], mitochondrial	3.40	1.66	1.66 × 10^−4^
METAP2	Methionine aminopeptidase 2	2.70	1.64	2.10 × 10^−4^
BCAR1	Breast cancer anti-estrogen resistance protein 1	2.79	1.62	3.33 × 10^−4^
AEBP1	Adipocyte enhancer-binding protein 1	4.84	1.60	6.06 × 10^−6^
TM9SF2	Transmembrane 9 superfamily member 2	3.64	1.59	7.08 × 10^−5^
ALDH3A1	Aldehyde dehydrogenase, dimeric NADP-preferring	30.54	1.59	3.96 × 10^−6^
EIF1	Eukaryotic translation initiation factor 1	3.51	1.58	5.59 × 10^−4^
FAM160B1	Protein FAM160B1	2.62	1.56	4.92 × 10^−4^
MAPRE2	Microtubule-associated protein RP/EB family member 2	3.10	1.55	1.37 × 10^−3^
ITGA2	Integrin alpha-2	61.27	1.53	2.81 × 10^−5^
EPS15	Epidermal growth factor receptor substrate 15	3.45	1.51	1.20 × 10^−3^
TCEB2; ELOB	Elongin-B	2.68	1.50	2.54 × 10^−3^
SUB1	Activated RNA polymerase II transcriptional coactivator p15	6.69	1.49	4.80 × 10^−4^
GAS6	Growth arrest-specific protein 6	3.57	1.49	9.31 × 10^−4^
LEPROT	Leptin receptor gene-related protein	4.86	1.48	1.21 × 10^−3^
PRCP	lysosomal Pro-X carboxypeptidase	3.25	1.47	7.48 × 10^−4^
TSG101	tumor susceptibility gene 101 protein	4.34	1.47	1.31 × 10^−3^
SLC1A5	Neutral amino acid transporter B (0)	22.90	1.46	2.29 × 10^−4^
FAM21C; WASHC2C	WASH complex subunit 2C	3.93	1.46	4.97 × 10^−3^
CTBP2	c-terminal-binding protein 2	4.67	1.46	5.77 × 10^−3^
ACTR1B	Beta-centractin	14.01	1.45	8.24 × 10^−4^
EXOC5	Exocyst complex component 5	2.62	1.45	4.09 × 10^−3^
UBE2K	Ubiquitin-conjugating enzyme E2 K	5.00	1.45	2.27 × 10^−3^
SH3PXD2B	SH3 and P.X. domain-containing protein 2B	5.85	1.45	1.77 × 10^−3^
SYNCRIP	Heterogeneous nuclear ribonucleoprotein Q	34.56	1.45	3.65 × 10^−4^
RETSAT	all-trans-retinol 13,14-reductase	3.86	1.44	7.73 × 10^−3^
TNC	Isoform 4 of Tenascin	137.79	0.71	7.03 × 10^−4^
TPM1	Isoform 3 of Tropomyosin alpha-1 chain	42.56	0.70	6.01 × 10^−6^
PPAP2B; PLPP3	Phospholipid phosphatase 3	14.13	0.69	4.17 × 10^−6^
MPST	3-mercaptopyruvate sulfurtransferase	3.58	0.69	6.19 × 10^−4^
HSPG2	Basement membrane-specific heparan sulfate proteoglycan core protein	22.22	0.68	2.86 × 10^−5^
TPM2	Isoform 2 of Tropomyosin beta chain	59.61	0.68	1.96 × 10^−6^
PAFAH1B2	platelet-activating factor acetylhydrolase I.B. subunit beta	3.24	0.68	1.06 × 10^−4^
HLA-C	HLA class I histocompatibility antigen, Cw-6 alpha chain	30.81	0.65	5.63 × 10^−5^
KDELR1	ER lumen protein-retaining receptor 1	3.22	0.62	1.58 × 10^−5^
P4HA2	Prolyl 4-hydroxylase subunit alpha-2	25.64	0.61	3.40 × 10^−9^
TRA2B	Transformer-2 protein homolog beta	9.60	0.61	1.90 × 10^−6^
HIST2H2AC	Histone H2A type 2-C	19.00	0.60	1.01 × 10^−9^
**NDUFA10**	NADH dehydrogenase [ubiquinone] 1 alpha subcomplex subunit 10, mitochondrial	2.71	0.59	4.62 × 10^−6^
NHP2	H/ACA ribonucleoprotein complex subunit 2	2.80	0.58	2.24 × 10^−7^
**COL12A1**	Collagen alpha-1 (XII) chain	97.95	0.57	9.99 × 10^−12^
HIST3H2A	Histone H2A type 3	18.81	0.52	7.02 × 10^−11^
RPS24	40S ribosomal protein S24	3.17	0.51	2.25 × 10^−9^
**COL1A2**	Collagen alpha-2 (I) chain	87.06	0.49	1.00 × 10^−17^
TAF15	TATA-binding protein-associated factor 2N	11.93	0.47	6.74 × 10^−11^
TUBA1A	Tubulin alpha-1A chain	188.76	0.41	2.22 × 10^−16^
SLC35E1	Solute carrier family 35 member E1	3.85	0.40	4.44 × 10^−16^
**COL1A1**	Collagen alpha-1 (I) chain	76.45	0.40	1.00 × 10^−17^
ANKMY1	Ankyrin repeat and MYND domain-containing protein 1	2.66	0.07	1.00 × 10^−17^
SLC39A7	Zinc transporter SLC39A7	4.18	0.01	1.00 × 10^−17^

^(1)^ The sum PEP score corresponds to the score calculated based on the posterior error probability (PEP) values of the peptide spectrum matches (PSM). The PEP indicates the probability that an observed PSM was a random event. The Sum PEP score was calculated as the negative logarithms of the PEP values of the connected PSM. Proteins significantly deregulated in HGF-1 cells after FcPs treatment are highlighted in bold. ^(2)^ Protein expressed only in cells treated with FcPs, but not in control cells.

**Table 4 molecules-29-03628-t004:** Comparison of total Carbon (C), Hydrogen (H), Nitrogen (N), Sulfur (S), and protein content obtained from the biomass species mushrooms.

	Carbon (%)	Hydrogen (%)	Nitrogen (%)	Sulfur (%)	Protein (Dry Weight, D.W., %)
*Fomitiporia chilensis*	40.03	6.08	2.14	0.97	18.24
*Pleurotus eus*	41.08	6.25	4.99	2.79	-
*Pleurotus ostreatus*	39.7–41.7	6.7–6.8	1.6–4.4	0.1–0.2	17.06
*Agaricus brasiliensis (fruiting body)*	34.42	5.56	2.50	0	28.9–39.2
*Agaricus brasiliensis (mycelial)*	32.07	4.36	1.63	0	-

## Data Availability

Data are contained within the article. The mass spectrometry proteomics data have been deposited to the ProteomeXchange Consortium via the PRIDE partner repository with the dataset identifier PXD048361 [83].

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
