# Peer review of "Immunomodulatory, Antioxidant, and Potential Anticancer Activity of the Polysaccharides of the Fungus Fomitiporia chilensis"

_molecules, 2024, doi:10.3390/molecules29153628_

Round 1
Reviewer 1 Report
Comments and Suggestions for Authors
After reading the article Immunomodulatory, antioxidant, and potential anticancer activity of the polysaccharides of the fungus Fomitiporia chilensis where shows the antioxidant, anti-cancer, and immunomodulatory activity of acidic polysaccharides obtained from the fungus Fomitiporia chilensis. I found the results very interesting for publication. However, I have some suggestions.
1. Authors must attach a conclusion in the abstract section
2. In the introduction section, finish with the objective of the research and eliminate line 85-90 since they are results.
3. Put the figures in better resolution.
4. Figure 4 homogenizes the title of the Y axis, for example Survival (%).
5. In figure 6, indicate that there is a significant difference in the figure caption.
6. Homogenize the subtitles throughout the writing.
7. In my opinion they have to restructure the discussion since the authors detail the results in many ways and that has to be in results and not in discussion, since it is very confusing.
8. In the reference part, italicize the abbreviated name of the journal.
Comments on the Quality of English Language
Minor editing of English language required
Author Response
For research article
Immunomodulatory, antioxidant, and potential anticancer activity of the polysaccharides of the fungus Fomitiporia chilensis.
|
Response to Reviewer 1 Comments |
||
|
1. Summary |
|
|
|
Thank you for addressing the reviewer’s comments. The revised manuscript is now ready for further consideration. We appreciate your attention to detail and your commitment to improving the quality of the article. Best regards, Claudia Pérez M. Corresponding author |
||
|
2. Questions for General Evaluation |
Reviewer’s Evaluation |
Response and Revisions |
|
Does the introduction provide sufficient background and include all relevant references? |
Yes/Can be improved/Must be improved/Not applicable |
The introduction has been modified. In this section the research objective has been included for clarity. |
|
Are all the cited references relevant to the research? |
Yes/Can be improved/Must be improved/Not applicable |
24 references have been added to strengthen the discussion of the results. |
|
Is the research design appropriate? |
Yes/Can be improved/Must be improved/Not applicable |
------------------ |
|
Are the methods adequately described? |
Yes/Can be improved/Must be improved/Not applicable |
------------------ |
|
Are the results clearly presented? |
Yes/Can be improved/Must be improved/Not applicable |
The results were reviewed and changes were made to improve the quality of the Figures (1, 3, 4, 5). |
|
Are the conclusions supported by the results? |
Yes/Can be improved/Must be improved/Not applicable |
------------------- |
|
3. Point-by-point response to Comments and Suggestions for Authors |
||
|
Comments 1: Authors must attach a conclusion in the abstract section |
||
|
Response 1: The authors agree with the suggestions. The conclusions were added (Page 2, Abstract section, Lines 45-49). “[The study on acidic polysaccharides from F. chilensis has unveiled their diverse biological activities, including antioxidant, anticancer, and immunomodulatory effects. These findings underscore the promising therapeutic applications of acidic polysaccharides from F. chilensis, warranting further pharmaceutical and medicinal research exploration.]” |
||
|
Comments 2: [In the introduction section, finish with the objective of the research and eliminate line 85-90 since they are results.] |
||
|
Response 2: The authors agree with the suggestions. Lines 85-90 were deleted. The objective of the research was added (Page 3, Introduction section, Lines 90-98) “[Fomitiporia species exhibit a wide range of biological activities, from cytotoxicity to antioxidant and immunomodulatory effects, highlighting their potential for medicinal uses. Understanding the mechanisms behind these activities could lead to developing new therapeutic interventions using the bioactive compounds found in these fungi. This research aimed to determine the antioxidant, anticancer, and immunomodulatory activity of acidic polysaccharides obtained from the fungus F. chilensis. Through a comprehensive analysis involving various assays and studies, the objective is to evaluate the therapeutic potential of these acidic polysaccharides and contribute to understanding their biological effects, particularly in the context of potential medicinal applications]” Comments 3: [Put the figures in better resolution.] Response 3: The authors agree with the suggestions. The figures were edited with a better resolution. Figure 1 was replaced (Page 4, Results section, Line 119-121). Figure 4 was replaced (Page 6, Results section, Line 165) Figure 5 was replaced (Page 7, section Results, Line 185) Figure 6 was replaced (Page 8, section Results, Line 207) Comments 4: [Figure 4 homogenizes the title of the Y axis, for example Survival (%). Response 4: The authors agree with the suggestions. In response to your comment about homogenizing the Y-axis title in Figure 4, I have made the necessary adjustments: In Figure 4 each graph now features a uniform Y-axis title, specifically "Survival (%)". You can find the changes on (Page 6, section Results, Line 165) Comments 5: [In figure 6, indicate that there is a significant difference in the figure caption] Response 5: The authors agree with the suggestions. In response to your comment on Figure 6, a necessary modification has been made to the legend. Indeed, the legend in Figure 6 now includes an explicit indication that there is a significant difference. In cases where there is a significant difference, these are indicated by an asterisk in the graph. “[Effect of FcPs on the HCT166 Cell Cycle. (A) Representative cell cycle histograms for HCT166 control conditions or treated with the polysaccharide at the pointed concentrations, measured after 16 h by propidium iodide staining and flow cytometry. (B) The quantitative analysis for the four cell cycle subpopulations in control and treated conditions. Data are means ± S.D. of at least three independent experiments (*p < 0.05, **p < 0.01)]” You can find the changes on (Page 8, section Results, Line 207-210) Comments 6: [Homogenize the subtitles throughout the writing.] Response 2: We appreciate the editor's comment on the homogenization of subtitles. In response to your comment, we have reviewed and standardized throughout the manuscript the subheadings throughout the text. In addition, we have ensured consistent formatting (capitalization, lowercase, title style, italics, superscript and subscript, etc.) and that key terms are repeated uniformly, and emphasis has been placed on the use of italics for the names of fungal species mentioned in the manuscript. Comments 7: [In my opinion they have to restructure the discussion since the authors detail the results in many ways and that has to be in results and not in discussion, since it is very confusing.] Response 7: We appreciate the reviewer's comment on the structure of the discussion section. We believe that clarity in the presentation of the results and their interpretation is essential for them to follow the thread of the study. To address this point, we restructured the discussion section. In this way, the results are presented more concisely in the results section and the discussion is reserved for interpreting and contextualizing these results. This should help avoid confusion and bring clarity to the article. You can find the changes on (Page 8, section Discussion, Line 262-557) Comments 8: [In the reference part, italicize the abbreviated name of the journal.] Response 8: We appreciate the editor's comment on the format of the references. Following the citation rules, I have made the necessary modifications in the references section. All references have been reviewed and updated to the requested format. You can find the changes on (Page 26, section References 1-84, Line 845-1065). |
||
|
4. Response to Comments on the Quality of English Language |
||
|
Point 1: Minor editing of English language required |
||
|
Response 1: The authors have carefully reviewed the manuscript and looked for grammatical errors, confusing phrases, or sentence structures that could be improved. With special attention to verb agreement, use of prepositions, and word choice. We have used grammar correction tools, such as Grammarly, to identify possible problems. |
||
|
5. Additional clarifications |
||
|
Dear reviewer, the authors have received comments and suggestions from two other reviewers. All suggestions have been considered and the manuscript has been extensively revised. In particular, the Discussion of Results section has been modified and divided into each of the experimental sections. In the manuscript, 24 relevant scientific references from the field have been added to support our results and conclusions. |
||

Reviewer 2 Report
Comments and Suggestions for Authors
1. Keywords
- add “Fomitiporia chilensis”
2. Materials and methods
- Lines 459 and 502: “F. chilensis”, italicized
- Lines 578, 586: “CO2” change to “CO2”
3. Results and discussion
- Line 105: “2,14%” correct to “2.14”
- The FT-IR spectra need an interpretation of the relevant signals related to polysaccharides.
- For the L.C.–HRMS Sugar Analysis, include information about the molecular weight and fragments of galactose, galacturonic acid, and glucose isomers (both α and β forms)
- “IC50” correct to “IC50”
- “E. coli” correct to “E. coli”
- Figure 1 and 6: should be adjusted clarity and resolution
4. Discussion
- “Ganoderma lucidum” correct to “Ganoderma lucidum”
- “Pleurotus ostreatus” correct to “Pleurotus ostreatus”
- In lines 314-315, where it mentions “and other minority monosaccharides,” provide additional details about these minor monosaccharides
- “Fomitiporia punctate” correct to “Fomitiporia punctate”
- “Gymnopilus imperialis” correct to “Gymnopilus imperialis”
- Line 336: “Pleurotus ostreatus and Ganoderma lucidum” correct to “P. ostreatus and G. lucidum”
Comments on the Quality of English Language
- Graphical Abstract is highly recommended.
- There seems to be a typographical error that needs correction.
Author Response
For research article
Immunomodulatory, antioxidant, and potential anticancer activity of the polysaccharides of the fungus Fomitiporia chilensis.
|
Response to Reviewer 2 Comments |
||
|
1. Summary |
|
|
|
Thank you for addressing the reviewer’s comments. The revised manuscript is now ready for further consideration. We appreciate your attention to detail and your commitment to improving the quality of the article. Best regards, Claudia Pérez M. Corresponding author |
||
|
2. Questions for General Evaluation |
Reviewer’s Evaluation |
Response and Revisions |
|
Does the introduction provide sufficient background and include all relevant references? |
Yes/Can be improved/Must be improved/Not applicable |
The introduction has been modified. In this section, the research objective has been included for clarity. |
|
Are all the cited references relevant to the research? |
Yes/Can be improved/Must be improved/Not applicable |
24 references have been added to strengthen the discussion of the results. |
|
Is the research design appropriate? |
Yes/Can be improved/Must be improved/Not applicable |
------------------ |
|
Are the methods adequately described? |
Yes/Can be improved/Must be improved/Not applicable |
------------------ |
|
Are the results clearly presented? |
Yes/Can be improved/Must be improved/Not applicable |
The results were reviewed and changes were made to improve the quality of the Figures (1, 3, 4, 5). |
|
Are the conclusions supported by the results? |
Yes/Can be improved/Must be improved/Not applicable |
------------------- |
|
3. Point-by-point response to Comments and Suggestions for Authors |
||
|
Comments 1: Keywords - add “Fomitiporia chilensis” |
||
|
Response 1: The authors agree with the suggestions. Fomitiporia chilensis was added in the Keywords (Page 2, Abstract section, Lines 52). |
||
|
Comments 2: 2. Materials and methods - Lines 459 and 502: “F. chilensis”, italicized - Lines 578, 586: “CO2” change to “CO2” |
||
|
Response 2: The authors agree with the suggestions. Fomitiporia chilensis was written in italics (Page 20, section 4.2. Collection, Authentication of Basidiocarps and Mycelial Culture, Lines 571) CO2 was correctly written in subscript (Page 21, section 4.4 Total Carbon (C), Hydrogen (H), Nitrogen (N), and Sulfur (S), and Protein content., Lines 618) Comments 3: Results and discussion a. Line 105: “2,14%” correct to “2.14” b. The FT-IR spectra need an interpretation of the relevant signals related to polysaccharides. c. For the L.C.–HRMS Sugar Analysis, include information about the molecular weight and fragments of galactose, galacturonic acid, and glucose isomers (both α and β forms) d. “IC50” correct to “IC50” e. “E. coli” correct to “E. coli” f. Figure 1 and 6: should be adjusted clarity and resolution Response 3: The authors agree with the suggestions. The figures were edited with a better resolution. a. The value "2.14%" was changed to "2.14" to ensure correct decimal notation. (Page 3, Results section, Line 112, and Table 1). b. Interpretation of FT-IR Spectra. In the results and discussion section, a more detailed interpretation of the relevant signals in the FT-IR spectra related to polysaccharides was provided. [The FT-IR spectra in the wavenumber range of 400 cm-1 to 4,000 cm-1 were utilized for data analysis. The FT-IR data indicated that the polysaccharides exhibit typical carbohydrate patterns, allowing for the identification of specific functional groups and molecular arrangements (Figure 1)]. (Page 4, Results section 2.1.2 Fourier Transform Infrared Spectroscopy (FT-IR), Line 119-121).] In section 3.1.2 Fourier Transform Infrared Spectroscopy (FT-IR), it is possible to find a more detailed description of FTIR spectra of FcPs, as well as the relevance for the characterization of polysaccharides and their bioactivity(Page Discussion section 2.1.2 Fourier Transform Infrared Spectroscopy (FT-IR), Lines 330-363) c. Information on L.C.-HRMS Sugar Analysis: -The analysis was done with GCMS so the title was changed. Concerning your comment, details on molecular weight and fragments of galactose, galacturonic acid, and glucose were added in the GCMS sugar analysis section. (Page 16, section 3.1.3 Gas chromatography-mass spectrometry (GC-MS), Line 378-383). [The derivative glucose, galactose, and galacturonic acid have a molecular weight of 482, 482, and 496 gmol-1 respectively. These monosaccharides present the same fragments 73, 133, 147, 204, 217 (m/z) when ionized by electronic impact at 70 eV so they can only be identified by comparing their retention times with those of their respective standards. Minor derived monosaccharides would have molecular masses of 380 gmol-1 for ribose and xylose and 394 for rhamnose and fucose, respectively] d. Suggested corrections for "IC50" (Pages 17 and 18, Results 3.2.2 Cell Viability of Lines RAW 264.7, U-937, HTC-116, and HGF-1 section, Line 433, 434, 435, 436 and 446, Lines 433, 434, 435, 436 and 438), and "E. coli", (Page 6, Results section, Line 175) were made. e. Suggested corrections for Figure 1 and 6: should be adjusted clarity and resolution Both Figures (1 and 6) have been changed. In addition, the figures 4,5 were also improved. Figure 1 was replaced (Page 4, Results section, Line 119-121). Figure 4 was replaced (Page 6, Results section, Line 165) Figure 5 was replaced (Page 7, section Results, Line 185) Figure 6 was replaced (Page 8, section Results, Line 207) Comments 4: Discussion a. “Ganoderma lucidum” correct to “Ganoderma lucidum” b. “Pleurotus ostreatus” correct to “Pleurotus ostreatus” c. In lines 314-315, where it mentions “and other minority monosaccharides,” provide additional details about these minor monosaccharides d. “Fomitiporia punctate” correct to “Fomitiporia punctate” e. “Gymnopilus imperialis” correct to “Gymnopilus imperialis” f. Line 336: “Pleurotus ostreatus and Ganoderma lucidum” correct to “P. ostreatus and G. lucidum” Response 4: We appreciate the editor's comment. In response to your comment, we have reviewed and standardized throughout the manuscript the subheadings throughout the text. In addition, we have ensured consistent formatting (capitalization, lowercase, title style, italics, superscript and subscript, etc.) and that key terms are repeated uniformly, and emphasis has been placed on the use of italics for the names of fungal species mentioned in the manuscript. a. “Ganoderma lucidum” (Page 14, Discussion section, Line 270; Page 17, Discussion section, Lines 396 and 398; Page 26, References section, Lines 871 and 875) b. “Pleurotus ostreatus” (Pages 15 y 17, Discussion section, Line 304, 396 and 398; Pages 26 and 27, References section, Lines 887 and 893) c. In lines 314-315, where it mentions “and other minority monosaccharides,” provide additional details about these minor monosaccharides. Thank you for your comment. Information on monosaccharides has been provided in the relevant section. [ Minor derived monosaccharides would have molecular masses of 380 gmol-1 for ribose and xylose and 394 for rhamnose and fucose, respectively](Page 16, 3.1.3 Gas chromatography-mass spectrometry (GC-MS), Line 381-382) d. “Fomitiporia punctate” (Page 3, Introduction section, Line 82; Page 16, Discussion section, Lines 383 Page 26 and 27, References section, Lines 863 and 931) e. “Gymnopilus imperialis” (Page 17, Discussion section, Line 389; Page 27, References section, Line 935) f. “Pleurotus ostreatus and Ganoderma lucidum” (Page 17, Discussion section, Line 396 y 398) Comments 5: Graphical Abstract is highly recommended. Response 5: In the submission of the manuscript the graphical abstract was provided. Comments 6: There seems to be a typographical error that needs correction. Response 2: The authors appreciate their thorough review. The entire manuscript with attention to identifying possible typographical errors. Pay special attention to numbers, symbols, abbreviations, and scientific terms. |
||
|
4. Response to Comments on the Quality of English Language |
||
|
Point 1: Minor editing of English language required |
||
|
Response 1: The authors have carefully reviewed the manuscript and looked for grammatical errors, confusing phrases, or sentence structures that could be improved. With special attention to verb agreement, use of prepositions, and word choice. We have used grammar correction tools, such as Grammarly, to identify possible problems. |
||
|
5. Additional clarifications |
||
|
Dear reviewer, the authors have received comments and suggestions from two other reviewers. All suggestions have been considered and the manuscript has been extensively revised. In particular, the Discussion of Results section has been modified and divided into each of the experimental sections. In the manuscript, 24 relevant scientific references from the field have been added to support our results and conclusions. |
||

Reviewer 3 Report
Comments and Suggestions for Authors
his manuscript mainly provided the chemical analysis and the antioxidant, anti-cancer, and immunomodulatory activities of the acidic polysaccharides obtained from the fungus Fomitiporia chilensis, while cell cycle and proteomic analysis suggested the possible molecular mechanisms for their biological activity. This work would provide well support for the development of Fomitiporia chilensis, especially their applications in medical and health industry. However, there are still many aspects required improved and perfected in the manuscript.
1. There are so many formatting and grammatical mistakes in the manuscript, please recheck the language and details carefully again.
All Latin names require italics, such as "E. coli" in line 171, "Ganoderma lucidum" in line 263, "Ganoderma and Pleurotus" in line 273, "Fomitiporia punctata" in line 322," Grifola frondose" in line 338, "F. chilensis" in line 344, "Fomitiporia chilensis" in line 690.
Superscripts or subscripts should be used correctly, such as "IC50", "µg mL-1", "pg mL−1" in section 2.2.3/2.2.4, "CO2" in line 578, "5 × 104 cells well-1" in line 614.
Please unify valid numbers, especially in the same experiment, such as"6.5 ± 0.3 and 6.99 ± 0.5 %" in line 140, " 1 mg mL-1 and 2.0 mg mL-1 " in lines 369-370.
Some uses of tense such as "induce" in line 41,"exhibit" in line 301 are wrong.
The typeface in some cases is "Time Newman" like that in line 203/211/563/583, is that right?
Line 117, please check the word "fractions" should be "fraction"?
Line 213, please check "values [of the peptide spectrum matches (PSM)".
Line 345, please check the word "activities" should be "activity"?
Line 567, " ...RAW 264.7." should be " ...RAW 264.7,...".
.Line 583," ...and (HGF-1)..." should be " ...and HGF-1..."?
2. Line 104, the full name for FcPs should be provided. Else, should this abbreviated word be italicized or non-italicized? Please confirm and unify the description in the manuscript.
3. What's the full name of "D.W." in Table 1?
4. In Figure 5b, the data on the 1.5 mg mL-1 polysaccharide is missing, and the S.D. on the 2.0 mg mL-1 polysaccharide is so large, could you conduct the experiment again to provide additional data and reduce the error? In addition, these data do not seem to have a clear concentration-activity trend, Please carry out some analysis in the discussion section.
5.In Figure 6, the p for the G2/M phase is > 0.05, referring that it is statistically insignificant, so, is the description in line 195 appropriate?
6. Line 214, please check the sentence "The Sum PEP score was calculated as the negative logarithms of the PEP values of the connected PSM".
7. The discussion section is very rich in content and poor in logical coherence, could you divide it into sections for better understanding?
8. The elemental contents of C, H, N, and S in fungal polysaccharides from different sources were showed in lines 256-299, can you explain the significance of their contents? It is recommended to provide a table for a more intuitive comparison.
9. Like question 9, in lines 312-331, the authors listed the monosaccharide compositions of many fungi-derived polysaccharides, please analyze and form a certain analytical summary. In line P332-342, the data should also be analyzed, not simply listed.
Comments on the Quality of English Language
No.
Author Response
For research article
Immunomodulatory, antioxidant, and potential anticancer activity of the polysaccharides of the fungus Fomitiporia chilensis.
|
Response to Reviewer 3 Comments |
||
|
1. Summary |
|
|
|
Thank you for addressing the reviewer’s comments. The revised manuscript is now ready for further consideration. We appreciate your attention to detail and your commitment to improving the quality of the article. Best regards, Claudia Pérez M. Corresponding author |
||
|
2. Questions for General Evaluation |
Reviewer’s Evaluation |
Response and Revisions |
|
Does the introduction provide sufficient background and include all relevant references? |
Yes/Can be improved/Must be improved/Not applicable |
The introduction has been modified. In this section, the research objective has been included for clarity. |
|
Are all the cited references relevant to the research? |
Yes/Can be improved/Must be improved/Not applicable |
24 references have been added to strengthen the discussion of the results. |
|
Is the research design appropriate? |
Yes/Can be improved/Must be improved/Not applicable |
The authors appreciate the reviewer's suggestion. However, we consider that the experimental design has allowed us to fulfill the objective proposed in this research. The methods of analysis have been adequate. The experimental design has been previously applied in other equivalent studies by authors and researchers in the area. The comment will be considered in a future proposal. |
|
Are the methods adequately described? |
Yes/Can be improved/Must be improved/Not applicable |
------------------ |
|
Are the results clearly presented? |
Yes/Can be improved/Must be improved/Not applicable |
The results were reviewed and changes were made to improve the quality of the Figures (1, 3, 4, 5). |
|
Are the conclusions supported by the results? |
Yes/Can be improved/Must be improved/Not applicable |
------------------- |
|
3. Point-by-point response to Comments and Suggestions for Authors |
||
|
Comments 1: 1. There are so many formatting and grammatical mistakes in the manuscript, please recheck the language and details carefully again. All Latin names require italics, such as "E. coli" in line 171, "Ganoderma lucidum" in line 263, "Ganoderma and Pleurotus" in line 273, "Fomitiporia punctata" in line 322," Grifola frondose" in line 338, "F. chilensis" in line 344, "Fomitiporia chilensis" in line 690. · “Ganoderma lucidum” (Page 14, Discussion section, Line 270; Page 17, Discussion section, Lines 396 and 398; Page 26, References section, Lines 871 and 875) · “Pleurotus ostreatus” (Pages 15 y 17, Discussion section, Line 304, 396 and 398; Pages 26 and 27, References section, Lines 887 and 893) · “E. coli", (Page 6, Results section, Line 175) were made. · “Fomitiporia punctate” (Page 3, Introduction section, Line 82; Page 16, Discussion section, Lines 383 Page 26 and 27, References section, Lines 863 and 931) · “Gymnopilus imperialis” (Page 17, Discussion section, Line 389; Page 27, References section, Line 935) · “Pleurotus ostreatus and Ganoderma lucidum” (Page 17, Discussion section, Line 396 y 398) · Fomitiporia chilensis was written in italics (Page 20, section 4.2. Collection, Authentication of Basidiocarps and Mycelial Culture, Lines 571) a. Superscripts or subscripts should be used correctly, such as "IC50", "µg mL-1", "pg mL−1" in section 2.2.3/2.2.4, "CO2" in line 578, "5 × 104 cells well-1" in line 614. · Suggested corrections for "IC50" (Pages 17 and 18, Results 3.2.2 Cell Viability of Lines RAW 264.7, U-937, HTC-116, and HGF-1 section, Line 433, 434, 435, 436 and 446, Lines 433, 434, 435, 436 and 438). · CO2 was correctly written in subscript (Page 21, section 4.4 Total Carbon (C), Hydrogen (H), Nitrogen (N), and Sulfur (S), and Protein content., Lines 618) b. Please unify valid numbers, especially in the same experiment, such as"6.5 ± 0.3 and 6.99 ± 0.5 %" in line 140, " 1 mg mL-1 and 2.0 mg mL-1 " in lines 369-370. We verify that the superscript and subscript symbols are used correctly. Valid numbers were unified · "6.5 ± 0.3% and 6.99 ± 0.5 %"( Page 5, section2.2.1 Antioxidant Activity (ABTS method) Line 144 · " 1.0 mg mL-1 and 2.0 mg mL-1 " (Page 6, section Results Line 185; page 18,Discussion section, Line 481) c. Some uses of tense such as "induce" in line 41,"exhibit" in line 301 are wrong. “induce” was corrected (Page 5, Line 62) "exhibit" was corrected (Page 16, Line 352) d. The typeface in some cases is "Time Newman" like that in line 203/211/563/583, is that right? The typeface was corrected in the manuscript e. Line 117, please check the word "fractions" should be "fraction"? "fractions" was corrected (Page 15, Line 204) f. Line 213, please check "values [of the peptide spectrum matches (PSM)". “values [of the peptide spectrum matches (PSM)", was corrected (Page 12, Line 220) g. Line 345, please check the word "activities" should be "activity"? “Activities” was corrected by activity (Page 17, Line 423), h. Line 567, " ...RAW 264.7." should be " ...RAW 264.7,...". RAW 264.7 was corrected (Page 17, Line 430) i. Line 583," ...and (HGF-1)..." should be " ...and HGF-1..."? Parentheses have been deleted Page 17, Line 430 |
||
|
Comments 2: Line 104, the full name for FcPs should be provided. Else, should this abbreviated word be italicized or non-italicized? Please confirm and unify the description in the manuscript. Response 2: The authors agree with the suggestions. The full name for FcPs was provided. The abbreviation corrected throughout the un-italicized version of the manuscript. Fomitiporia chilensis polysaccharides (FcPs) (Page 5, Line 111) Comments 3: What's the full name of "D.W." in Table 1? Response 3: In name was provided in Table 1. (dry weight, D.W., %) page 3, Line 114 Comments 4: In Figure 5b, the data on the 1.5 mg mL-1 polysaccharide is missing, and the S.D. on the 2.0 mg mL-1 polysaccharide is so large, could you conduct the experiment again to provide additional data and reduce the error? In addition, these data do not seem to have a clear concentration-activity trend, Please carry out some analysis in the discussion section. Response 4: The researchers agree that the observed differences are significant and warrant another experiment replication. Despite differences between the three replicates, the TNF levels were notably higher. The data was thoroughly reviewed, and all data points were accounted for. We conducted serial dilutions starting from 2 mg mL-1 at a rate of 1:2. It's crucial to note that the concentrations differed in the two cell types as the compound's effect on the two cell types was not equal. The dose-response effect was not linear; the response began at 2 mg mL-1. The effect became noticeable in the raw samples starting at the third data point, and the response was already saturated. A much higher concentration of THP-1 was needed to observe a response. It's important to acknowledge that these are not the same cells or the same cytokine, so their effects cannot be directly compared. Furthermore, if the concentration in THP-1 were increased further, the difference would continue to grow until saturation. Unfortunately, we could not test further due to the limited availability of the F. chilensis polysaccharides (FcPs) and the potential cytotoxicity associated with increasing the dose. At the concentration used, there was a notable increase in apoptosis (Subg1 phase) and a significant decrease in G2/M, underscoring the need for caution in dose escalation due to potential cytotoxicity. Comments 5: In Figure 6, the p for the G2/M phase is > 0.05, referring that it is statistically insignificant, so, is the description in line 195 appropriate? Response 5: The untreated control serves as the baseline for comparing our treatments. The “positive” control (2ME) is used to ensure that the trial has been conducted properly and is not meant to be compared to the effect of F. chilensis polysaccharides (FcPs) or any other drug. The SubG1 population increases significantly with the dose, leading to a consistent decrease in other cell cycle phases. In phase G0/G1, there is a slight decrease at all concentrations, but it is insignificant, except at 1 mg mL-1. However, this effect is not consistently repeated with higher doses. Similar changes occur in phase S, with slight decreases at all three concentrations, but they are insignificant. On the other hand, the reduction in G2/M at the highest concentration tested is significant, indicating an impact on the cells entering cell division. In addition to increasing apoptosis at the lowest concentration, it also affects proliferation, with a significant difference compared to the untreated control at the highest concentration. To avoid confusion with the 2ME, the legend in Figure 6 was modified as follows; [ Figure 6. Effect of FcPs on the HCT166 Cell Cycle. (A) Representative cell cycle histograms for HCT166 control conditions or treated with the polysaccharide at the pointed concentrations, measured after 16 h by propidium iodide staining and flow cytometry. (B) The quantitative analysis for the four cell cycle subpopulations in control and treated conditions. Data are means ± S.D. of at least three independent experiments (*p < 0.05, **p < 0.01).] Comments 6: Line 214, please check the sentence "The Sum PEP score was calculated as the negative logarithms of the PEP values of the connected PSM". Response 6: For the authors, the sentence is correct Comments 7: The discussion section is very rich in content and poor in logical coherence, could you divide it into sections for better understanding? Response 7: We appreciate the reviewer's comment on the structure of the discussion section. We believe that clarity in the presentation of the results and their interpretation is essential for them to follow the thread of the study. To address this point, we restructured the discussion section. In this way, the results are presented more concisely in the results section and the discussion is reserved for interpreting and contextualizing these results. This should help avoid confusion and bring clarity to the article. You can find the changes on (Page 8, section Discussion, Line 262-557) Comments 8: The elemental contents of C, H, N, and S in fungal polysaccharides from different sources were showed in lines 256-299, can you explain the significance of their contents? It is recommended to provide a table for a more intuitive comparison. Response 8: The authors are grateful for the suggestion. The following has been added to the manuscript to explain the significance of its contents. In addition, a Table was added in the discussion to allow a better understanding of the data by comparing it with the other most known species of fungi. Table 4. Comparison of Total Carbon (C), Hydrogen (H), Nitrogen (N), Sulphur (S), and protein content obtained from the biomass species mushrooms. (Page 15, Line 328) The following was added to the manuscript to explain the significance of its contents; [The elemental composition of polysaccharides, encompassing carbon, hydrogen, nitrogen, and sulfur, is crucial for understanding their chemical structure and potential biological activities. Analyzing these elements provides valuable insights into the properties and functions of polysaccharides, aiding in establishing structure-function relationships and identifying possible applications in various fields, including medicine, nutrition, and bio-technology] References 13, 14. (Page 15, Line 294-299). 13. Mohammed, A.S.A.; Naveed, M.; Jost, N. Polysaccharides; Classification, Chemical Properties, and Future Perspective Applications in Fields of Pharmacology and Biological Medicine (A Review of Current Applications and Upcoming Poten-tialities). J. Polym. Environ. 2021, 29, 2359–2371, doi:10.1007/s10924-021-02052-2. 14. Rong, L.; Shen, M.; Zhang, Y.; Yu, H.; Xie, J. Food Polysaccharides and Proteins: Processing, Characterization, and Health Benefits. Foods 2024, 13, 1113, doi:10.3390/foods13071113 Comments 9: Like question 9, in lines 312-331, the authors listed the monosaccharide compositions of many fungi-derived polysaccharides, please analyze and form a certain analytical summary. In line P332-342, the data should also be analyzed, not simply listed. Response 9: Thank you for your comment. Additional Information has been provided in the relevant section. [The derivative glucose, galactose, and galacturonic acid have a molecular weight of 482, 482, and 496 gmol-1 respectively. These monosaccharides present the same fragments 73, 133, 147, 204, 217 (m/z) when ionized by electronic impact at 70 eV so they can only be identified by comparing their retention times with those of their respective standards. Mi-nor derived monosaccharides would have molecular masses of 380 gmol-1 for ribose and xylose and 394 for rhamnose and fucose, respectively.]Page 16, Line 379-383. [This contributes to the overall bioactivity of the polysaccharides extracted from these spe-cies. In conclusion, the monosaccharide compositions of mushroom polysaccharides, in-cluding the presence of specific monosaccharides and their ratios, are crucial factors that influence the bioactivity and potential health-promoting properties of these compounds. Understanding the monosaccharide profiles of FcPs is essential for elucidating their biological activities and exploring their therapeutic potential]. Page 17, Line 410-416 |
||
|
4. Response to Comments on the Quality of English Language |
||
|
Point 1: Minor editing of English language required |
||
|
Response 1: The authors have carefully reviewed the manuscript and looked for grammatical errors, confusing phrases, or sentence structures that could be improved. With special attention to verb agreement, use of prepositions, and word choice. We have used grammar correction tools, such as Grammarly, to identify possible problems. |
||
|
5. Additional clarifications |
||
|
Dear reviewer, the authors have received comments and suggestions from two other reviewers. All suggestions have been considered and the manuscript has been extensively revised. In particular, the Discussion of Results section has been modified and divided into each of the experimental sections. In the manuscript, 24 relevant scientific references from the field have been added to support our results and conclusions. |
||

Round 2
Reviewer 1 Report
Comments and Suggestions for Authors
The authors responded to the suggestions